# Numerically Evaluation of FRP-Strengthened Members under Dynamic Impact Loading

**Faham Tahmasebinia [1,\*], Linda Zhang [1], Sangwoo Park [1] and Samad Sepasgozar [2]**

[1] School of Civil Engineering, The University of Sydney, Sydney, NSW 2006, Australia; lzha4304@uni.sydney.edu.au (L.Z.); spar9387@uni.sydney.edu.au (S.P.)
[2] Faculty of Built Environment, The University of New South Wales, Sydney, NSW 2052, Australia; sepas@unsw.edu.au
\* Correspondence: Faham.tahmasebinia@sydney.edu.au

**Abstract:** Reinforced concrete (RC) members in critical structures, such as bridge piers, high-rise buildings, and offshore facilities, are vulnerable to impact loads throughout their service life. For example, vehicle collisions, accidental loading, or unpredicted attacks could occur. The numerical models presented in this paper are shown to adequately replicate the impact behaviour and damage process of fibre-reinforced polymer (FRP)-strengthened concrete-filled steel tube (CFST) columns and Reinforced Concrete slabs. Validated models are developed using Abaqus/Explicit by reproducing the results obtained from experimental testing on bare CFST and RC slab members. Parameters relating to the FRP and material components are investigated to determine the influence on structural behaviour. The innovative method of using the dissipated energy approach for structural evaluation provides an assessment of the effective use of FRP and material properties to enhance the dynamic response. The outcome of the evaluation, including the geometrical, material, and contact properties modelling, shows that there is an agreement between the numerical and experimental behaviour of the selected concrete members. The experimentation shows that the calibration of the models is a crucial task, which was considered and resulted in matching the force–displacement behaviour and achieving the same maximum impact force and displacement values. Different novel and complicated Finite Element Models were comprehensively developed. The developed numerical models could precisely predict both local and global structural responses in the different reinforced concrete members. The application of the current numerical techniques can be extended to design structural members where there are no reliable practical guidelines on both national and international levels.

**Keywords:** numerical modelling; reinforced concrete members; fibre-reinforced polymer; concrete-filled steel tube; dynamic simulations



## 1. Introduction

There is an increasing demand for building designers to improve the protection of significant infrastructure, as they are vulnerable to dynamic impact loading events. Understanding the structural response can minimise the risk to life and economic losses in catastrophic impact events. Applying fibre-reinforced polymer (FRP) sheets and strips is a popular method of external reinforcement to enhance impact resistance. Structural safety is improved due to its high strength-to-weight ratio and durability. Thorough experimental testing of FRP-strengthened specimens may not be possible due to difficulties replicating realistic high-velocity impact loading scenarios and limitations in facilities. Hence, there is still a significant knowledge gap in how FRP-strengthened elements will respond and the most efficient way to design the members. Numerical methods are a practical tool to understand this complex loading scenario.

The additional confinement force provided to circular concrete-filled steel tube (CFST) sections by externally bonded FRP wrapping was demonstrated to increase its ductility,

load-bearing, and energy absorption capacity [1–3]. FRP strips provide extra shear and flexural strength and crack resistance for slab members.

Numerical models developed in existing literature have identified the effect of various parameters on the structural response on the structural performance of FRP-strengthened members. Alam et al. [4] used Abaqus/Explicit to conduct a numerical study on FRP-strengthened CFST columns under transverse impact loading. The experimental tests performed by Chen et al. [5] were used to validate the numerical models. The validated models were used for parametric studies to observe the effects of several parameters on the responses of the impacted columns. Good agreement was found with the experimental results, confirming the ability of finite element (FE) simulations to accurately predict peak impact force, maximum lateral displacement, and FRP failure modes. However, due to the limited number of trials conducted, the contribution of the parameters cannot be fully confirmed. The only performance measure used in the parametric studies was maximum displacement.

Currently, there are no specific industry standards or design codes relating to complex dynamic impact loading conditions. Rudimentary guidance is given in the Australian Standards [6], Eurocode (EC) [7], AASHTO code [8], and American Concrete Institute standard [9] for the design of vehicle impacts on structural members. Transverse impacts are suggested to be treated as an equivalent static force (ESF) on the structure. This approximation has been questioned by various researchers [10–13], where it has been found to often be unsafe or overly conservative. Assessment of the structural performance may be estimated using the energy method approach. The energy dissipated by the member acts as a guide on how the ductility and deformability of a structure can be improved. Such information is crucial in evaluating the impact performance and adequacy of column members and provides predictions of post-failure behaviour [14–20].

There is a lack of detailed design guidance for FRP-strengthened structural members for dynamic impact loading conditions. The ability to find a mathematical relationship between the parameters extends the scope of the findings for practical application. Parametric studies with additional simulations are needed for quantitative measurements of performance and structural capacity.

## 2. Numerical Modelling

The column and slab models are created and validated using experimental studies from the literature. The Chen et al. [5] experiments are replicated for the CFST column and the Hrynyk and Vecchio [21] experiments for the reinforced concrete (RC) slab.

### 2.1. Material Modelling

#### 2.1.1. Concrete

The concrete core of CFST specimens is subjected to a laterally confining pressure provided by the steel tube. This enhances the uniaxial compressive strength and ductility of the concrete [22]. Therefore, the concrete core of the CFST specimens was modelled with consideration of the confinement effect from the steel tube section. The confined compressive stress of concrete, $f_{cc}$ is calculated from the unconfined cylindrical compressive stress, $f_c$ using the confined concrete model proposed by Mander et al. [23].

$$f_{cc} = f_c + k_1 f_l \qquad (1)$$

where the term $f_l$ represents the confining pressure created by the steel tube around the concrete core. The value of $f_l$ is derived from the empirical equation obtained by Hu et al. [22]:

$$\frac{f_l}{f_y} = 0.043646 - 0.000832\left(\frac{D}{t}\right) \text{ for } 21.7 \leq \frac{D}{t} \leq 47 \qquad (2)$$

The constant $k_1$ is obtained from experimental data, based on the increase in the ultimate strength. A value of 4.1 is adopted from the studies of Richart et al. [24]. Thus,

a value of 55.3 MPa for $f_{cc}$ was calculated from the 38.7 MPa value of $f_c$ reported in the Chen et al. [5] experiments. The Young's Modulus of the confined concrete, $E_{cc}$, is calculated using the empirical formula with reasonable accuracy [25]:

$$E_{cc} = 4700\sqrt{f_{cc}} \text{ MPa} \tag{3}$$

The concrete damage plasticity model provided in ABAQUS/Explicit was used to characterise the behaviour of concrete under impact loading. The plastic-damage constitutive model for plain concrete developed by Cicekli et al. [26] was adopted to account for the different responses under tensile and compressive loading [27–30].

### 2.1.2. Steel

The isotropic classic metal plasticity model is used to consider the elastic-plastic behaviour of the steel material used for the tube section and reinforcement in the slab. The yield stress and corresponding plastic strain are calculated from EC2 [31]. Based on the Cowper–Symonds power law input available on Abaqus/Explicit, a strain rate of multiplier factor of $40.4^{-1}$ and exponent of 5 is adopted [32].

### 2.1.3. Fibre-Reinforced Polymer (FRP)
#### Column

The unidirectional FRP sheet used in this study was assumed to have a limited plastic region. Damage is defined in Abaqus/Explicit to specify the material damage initiation criteria and associated damage evolution. The shear damage model is selected, which predicts the onset of damage due to shear band localization [33]. This deformation mechanism is a narrow region of plastic shearing strain, which is comparable to the phenomena experienced by ductile FRP materials subjected to localised impact forces [34]. The point of rupture is given by the fracture strain, which is assumed to be 0.05 [35]. The shear stress ratio was assumed to be 0.6, for a typical high-strength ductile metal [36].

#### Slab

The FRP was assumed to be linear elastic until failure. Lamina elastic properties and Hashin [37] damage parameters were defined for this fibre composite material. The elastic and Hashin [37] damage properties obtained for the CFRP from literature are summarised in Tables 1 and 2 below.

**Table 1.** Elastic lamina properties of CFRP used for slab

| $E_1$ / MPa | $E_2$ / MPa | $\nu_{12}$ | $G_{12}$ / MPa | $G_{13}$ / MPa | $G_{23}$ / MPa |
|---|---|---|---|---|---|
| 171,000 | 9000 | 0.342 | 4800 | 4800 | 4500 |

**Table 2.** Hashin Damage Properties [38].

| *XT* | *XC* | *YT* | *YC* | *SL* | *ST* |
|---|---|---|---|---|---|
| 2050 | 1500 | 74 | 190 | 81 | 81 |

Where *XT*: longitudinal tensile strength, *XC*: longitudinal compressive strength, *YT*: transverse tensile strength, *YC*: transverse compressive strength, *SL*: longitudinal shear strength, *ST*: transverse shear stress.

CFRPs often have compressive strengths two-thirds of tensile strengths and so the initial tensile and compressive fracture energies were initially set at 150 and 100 mJ/mm$^3$, respectively. Note that varying compression fracture energy did not influence the results in both longitudinal and transverse directions. The damage stabilisation parameter and the viscosity coefficient were assumed to be 0.0001 in all directions.

The composite layup was used to define the direction and thickness of the CFRP strips. The directions of non-intersecting strips were defined in their longitudinal direction.

The overlap at the intersection of the strips was modelled by having the top two layers in the x-direction and the bottom two layers in the y-direction.

Adhesive

For a simplified analysis, the behaviour and failure of interface elements is not included in the scope of this study. Previous numerical studies have shown the epoxy adhesive used for FRP-wrapping to further minimise lateral displacement [4]. Hence, the omission of the adhesive is deemed a conservative approach.

2.1.4. Summary of Material Properties

Material properties of the concrete, steel, and FRP used in the numerical models are summarised (Table 3). These are taken from the experimental values (Table 4).

**Table 3.** Material properties of the column.

| Component | Parameter | Value |
|-----------|-----------|-------|
| **Steel tube** | $\rho_{st}$ $(\mathrm{kg/m^3})$ | 7800 |
| | $E_{st}$ (GPa) | 210 |
| | $f_{y,st}$ (MPa) | 245 |
| | $v_{st}$ | 0.3 |
| **Concrete core** | $\rho_{conc}$ $(\mathrm{kg/m^3})$ | 2400 |
| | $E_{conc}$ (GPa) | 32.5 |
| | $f_{conc}$ (MPa) | 38.7 |
| **FRP** | $\rho_{FRP}$ $(\mathrm{kg/m^3})$ | 1600 |
| | $E_{FRP}$ (GPa) | 200 |
| | $v_{FRP}$ | 0.3 |
| **Steel general** | $\rho_{st,g}$ $(\mathrm{kg/m^3})$ | 7800 |
| | $E_{st,g}$ (GPa) | 200 |
| | $f_{y,st,g}$ (MPa) | 500 |
| | $v_{st,g}$ | 0.3 |

**Table 4.** Material properties of the slab.

| Component | Parameter | Value |
|-----------|-----------|-------|
| **Steel** | $\rho_{st}$ $(\mathrm{kg/m^3})$ | 7850 |
| | $E_{st}$ (GPa) | 193 |
| | $f_{y,st}$ (MPa) | 489 |
| | $f_u$ (MPa) | 597 |
| | $v_{st}$ | 0.3 |
| **Concrete** | $\rho_{conc}$ $(\mathrm{kg/m^3})$ | 2441 |
| | $E_{conc}$ (GPa) | 36.8 |
| | $f_{conc}$ (MPa) | 69.4 |
| **Steel general** | $\rho_{st,g}$ $(\mathrm{kg/m^3})$ | 7800 |
| | $E_{st,g}$ (GPa) | 200 |
| | $f_{y,st,g}$ (MPa) | 500 |
| | $v_{s,g}$ | 0.3 |

*2.2. Element Modelling*

2.2.1. Column

Solid C3D8R elements with one integration point are selected to model the components. A bare CFST specimen was developed and analysed using identical geometries, boundary conditions, and loading as detailed in the Chen et al. [5] experiments. An inner diameter of 107 mm for the concrete fill was used, with a steel tube wall thickness of 3.5 mm. The overall length of the member was 1700 mm, with simply supported boundary conditions. To represent the experimental loading configuration, the CFST model was subjected to an impact at mid-span using an 80 kg drop weight with a 150 mm hemispherical head. A 5.48 m/s impact velocity of the drop weight at the critical stage before contact was replicated. The FE model created to simulate the experiment set-up is shown in Figure 1.

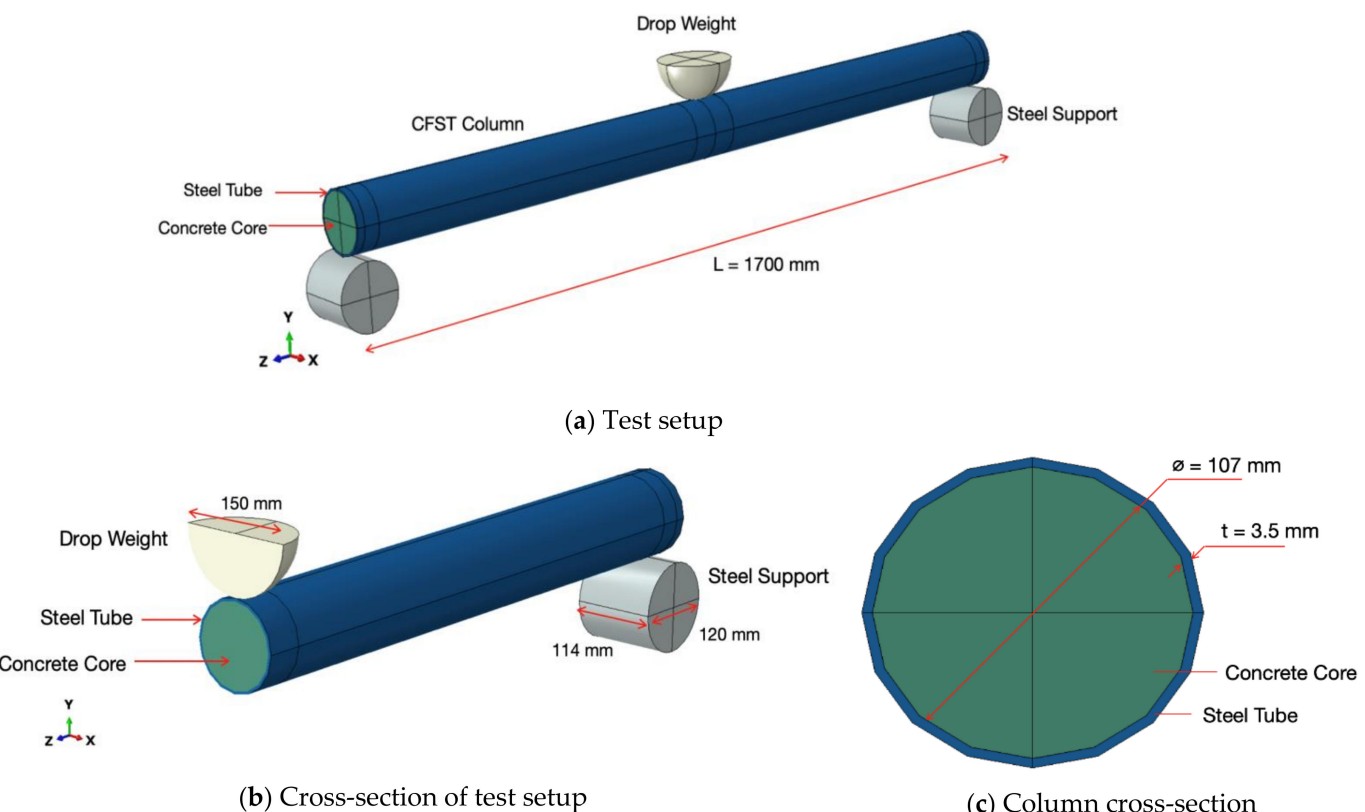

(**a**) Test setup

(**b**) Cross-section of test setup        (**c**) Column cross-section

**Figure 1.** Bare concrete-filled steel tube (CFST) column.

The contact pair algorithm on ABAQUS/Explicit is used to define the interactions between the components. Surface-to-surface contacts are established between engaging elements. The interface element for the steel tube and concrete core allows the surfaces to act discreetly under tensile forces [39]. Tangential and normal behaviour properties are defined. The friction coefficient is specified to account for the shear stress of the surface traction and contacting pressure [33]. A friction coefficient of 0.36 was adopted between the drop weight and steel tube, and 0.25 between the steel tube and concrete core. Previous studies have found these values to accurately represent experimental conditions [39,40].

The steel tube is assigned as the slave surface when it is in contact with the drop weight (master surface), and as the master surface when it is in contact with the surface of the concrete core (slave surface). This was also consistent with previous numerical studies with CFST models [41,42]. A tie constraint was applied between the surfaces of the steel tube and the supports. The stiffer support was selected as the master surface.

### 2.2.2. Slab

An RC slab specimen unstrengthened by FRP was developed and analysed using identical geometries, boundary conditions, and loading given in the Hrynyk and Vecchi [21] experiments. A 3-dimensional 8-node solid element (C3D8R) was used to model the concrete slab, steel drop weight and the steel supports. A 2-node linear beam element in space (B31) was used to model the steel reinforcement inside the slab. A 4-node shell element with reduced integration points (S4R) was used to model the CFRP strips.

The drop weight of radius 400 mm was modelled by utilising the revolution technique. Note that the drop weight used in the experiment had a square cross sectional area. However, using a non-rounded impact surface caused excessive deformations which could not be calibrated. Hence, the drop weight was modelled as a sphere with a similar cross-sectional area. The square slab and steel supported were modelled by utilising the extrusion technique. The 1800 mm square slab was 130 mm thick and the supports were assumed to be 120 mm by 120 mm by 25 mm. The steel reinforcement was to be placed on each face of the slab. Reinforcement in x- and y-direction were modelled with the rounded and cogged end, respectively. Note that reinforcement in both faces has been modelled for x-direction, whereas reinforcement in only one face has been modelled in y-direction as shown in Figure 2, respectively. The reinforcement in y-direction on the opposite face is to be modelled in the assembly module. Finally, two strips of CFRP were modelled in both x- and y-directions.

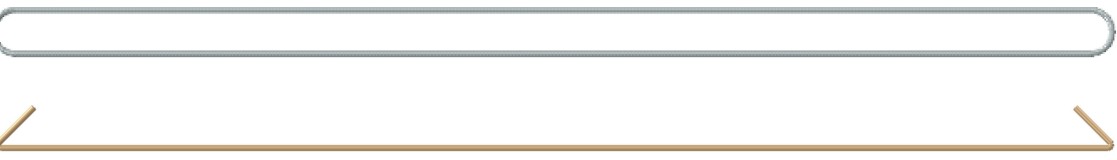

**Figure 2.** Reinforcement in (**top**) x- and (**bottom**) y-directions.

The dilation angle was initially assumed to be 30 degrees but later adjusted to 38 degrees by trial-and-error process during calibration. Mass scaling was set at 0.0001 for an initial time period 0.1.

Rigid body constraints were defined for the steel supports and the drop weight and the contact surfaces between the steel supports and the slab were tied to prevent any relative motion.

Surface-to-surface contact was defined between the drop weight and the slab. Normal and tangential behaviours with 0.3 friction coefficient were defined as the interaction properties. The CFRP strips were also tied to the slab but the relative motion in rotational degrees of freedom were allowed. Lastly, the steel reinforcements were embedded in the concrete slab.

The created parts were assembled as shown in Figure 3. The drop weight was positioned at the centre of the slab. The centre of the steel supports was positioned 145 mm from the edges and placed on top and bottom of the slab. The steel reinforcements were placed using a linear pattern tool with 130 mm spacing with 16 mm clear cover. The radial pattern tool was used for cogged-end y-direction steel reinforcement to obtain the reinforcement on the opposite face. The reinforcement at the top and bottom were rotated by 5 degrees in the opposite direction to avoid overlapping. The centreline of each CFRP strip was positioned 100 mm from the centre of the slab. The impact velocity of 8000 mm/s was applied as a boundary condition at step 1 for the drop weight.

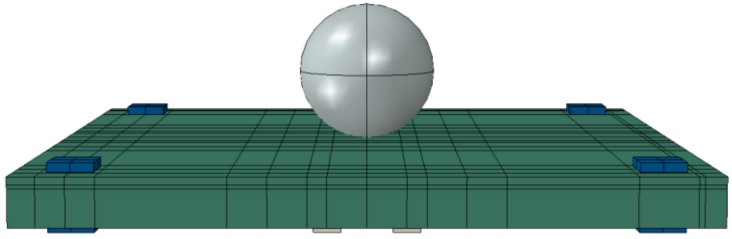

**Figure 3.** Assembly of parts.

*2.3. Model Validation*

Prior to performing the FE parametric studies, the accuracy of the numerical models and assumptions was verified. The adequacy of the adopted constitutive material properties and modelling strategies is validated against available test results. The accuracy of the models was assessed qualitatively (i.e., by comparing the general shape force–displacement curve) and quantitatively (i.e., maximum load and displacement values given in Table 5).

**Table 5.** Experimental maximum force and displacement values for column [43], and slab [21].

| *Member* | *Maximum Force, $F_{exp}$* (**kN**) | *Maximum Displacement, $d_{exp}$* (**mm**) |
|---|---|---|
| *Column* | *456.2* | *11* |
| *Slab* | *556.0* | *13* |

2.3.1. Mesh Refinement
Column

Initially, the same mesh size was used for both the steel tube and concrete core components. Table 6 gives details of a sample of the trials. The corresponding force–displacement curves are given in Figure 4.

**Table 6.** Mesh sizes for initial trials using the same steel and concrete mesh sizes.

| | **Case** | **A** | **B** | **C** | **D** | **E** |
|---|---|---|---|---|---|---|
| **Steel** | Global | 20 | 20 | 25 | 20 | 20 |
| | Local | 9 | 6 | 6 | 4 | 6 |
| **Concrete** | Global | 20 | 20 | 25 | 20 | 20 |
| | Local | 9 | 6 | 6 | 4 | 6 |
| | Weight | 11 | 11 | 11 | 10 | 10 |
| | Support | 11 | 11 | 11 | 10 | 10 |

Different element sizes of the steel tube and concrete core of the CFST were then tested. A finer mesh for the concrete slave surface was adopted to avoid penetration from the master steel surface.

Table 7 gives details of a selection of the trials where different element sizes of the steel tube and concrete core of the CFST were tested. The mesh used in trial E from Stage 1 is included for comparative purposes. Resulting force–displacement curves are given in Figure 5.

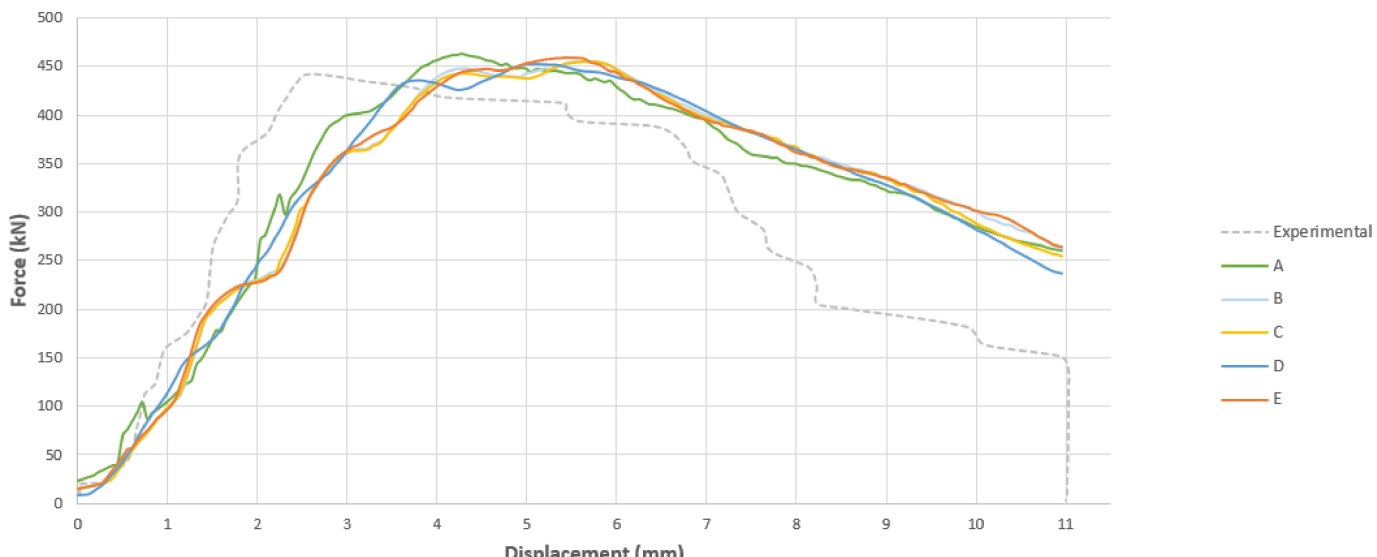

**Figure 4.** Force–displacement graph for initial trials using the same steel and concrete mesh sizes. Mesh Type A, B, C, D and E refer to the Table 6.

**Table 7.** Mesh sizes for trials using different element sizes for steel tube and concrete core.

| Material | Case | F | G | H | I |
|---|---|---|---|---|---|
| **Steel** | Global | 20 | 20 | 25 | 25 |
| | Local | 6 | 6 | 9 | 9 |
| **Concrete** | Global | 25 | 20 | 20 | 20 |
| | Local | 9 | 9 | 6 | 4 |
| | Weight | 10 | 10 | 10 | 10 |
| | Support | 10 | 10 | 10 | 10 |

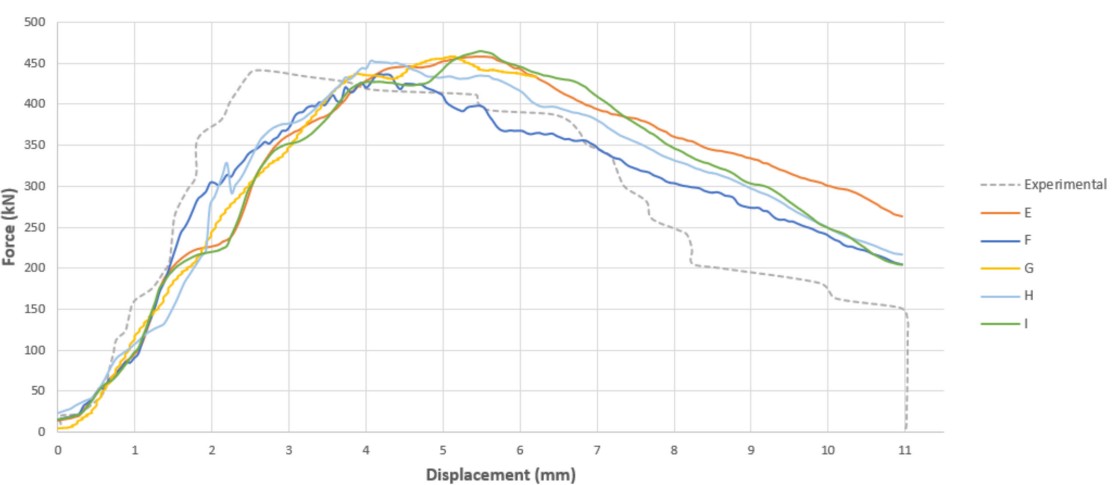

**Figure 5.** Force–displacement graph for trials using different element sizes for steel tube and concrete core. Mesh Type E, F, G, H and I refer to the Table 7.

The results of the maximum force and displacement values obtained for a selection of trials is given in Table 8 below. The time period is maintained at 0.002 s. These were compared to the values shown in Table 8.

**Table 8.** Comparison of experimental and numerical maximum force and displacement values.

| Trial No. | Mass Scaling ($\times 10^{-6}$) | F (kN) |
|:---:|:---:|:---:|
| 1 | 3.0 | 608.88 |
| 2 | 2.4 | 496.67 |
| 3 | 2.3 | 491.33 |
| 5 | 2.1 | 474.81 |
| 6 | 2.0 | 449.66 |

An initial stable time increment under $5.0 \times 10^{-6}$ was tested for each trial based on literature guidance [33]. The influence on $F_{FE}$ was observed, and was adjusted by increments of $0.1 \times 10^{-6}$ until values approached $F_{exp}$. The process is shown for Case A in Table 9.

**Table 9.** Mass scaling trials for Case A.

| Trial No. | Mass Scaling ($\times 10^{-6}$) | F (kN) |
|:---:|:---:|:---:|
| 1 | 3.0 | 608.88 |
| 2 | 2.4 | 496.67 |
| 3 | 2.3 | 491.33 |
| 5 | 2.1 | 474.81 |
| 6 | 2.0 | 449.66 |

An optimal mesh for the FE model was selected based on the closest matching with the experimental force–displacement curve. The results indicate how closer convergence to experimental behaviour may be achieved by adopting different mesh sizes for the CFST components. Closer matching to the experimental curve is achieved in Cases H and I. The computational time for Case H is less than half of that of Case I, yet its accuracy is not compromised. Hence, the meshing conditions used in trial H were selected.

Slab

The same mesh refinement was applied to the slab and steel reinforcement initially. The force–displacement graphs for mesh refinement of the slab and steel reinforcements are shown in Figure 6. The numbers in the legend show the seed sizes of the slab and the steel reinforcement, respectively. Seed sizes of 6 and 8 for the slab had the same force–displacement graph which suggested convergence. Hence, a seed size of 8 was selected for the slab.

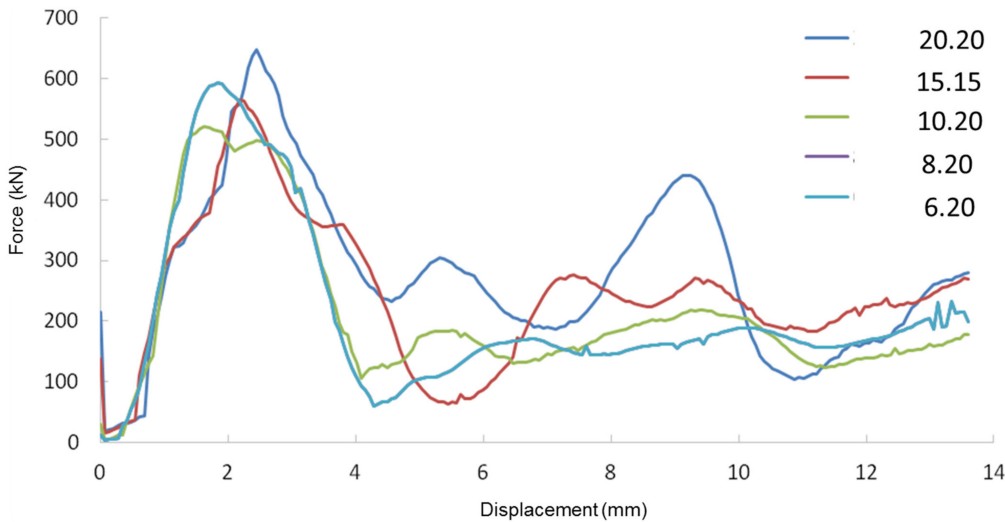

**Figure 6.** Forceversus displacement Figure in both slab and steel reinforcement mesh.

Local seeding in the impact region was then considered. The force–displacement graphs for local seeding mesh refinement of the slab are shown in Figure 7. The numbers in the legend show the seed sizes of the slab in the non-impact zone. No major differences were observed in the force–displacement graphs, and the computational time was also approximately the same. Thus, a seed size of 20 was chosen for non-impact region of the slab.

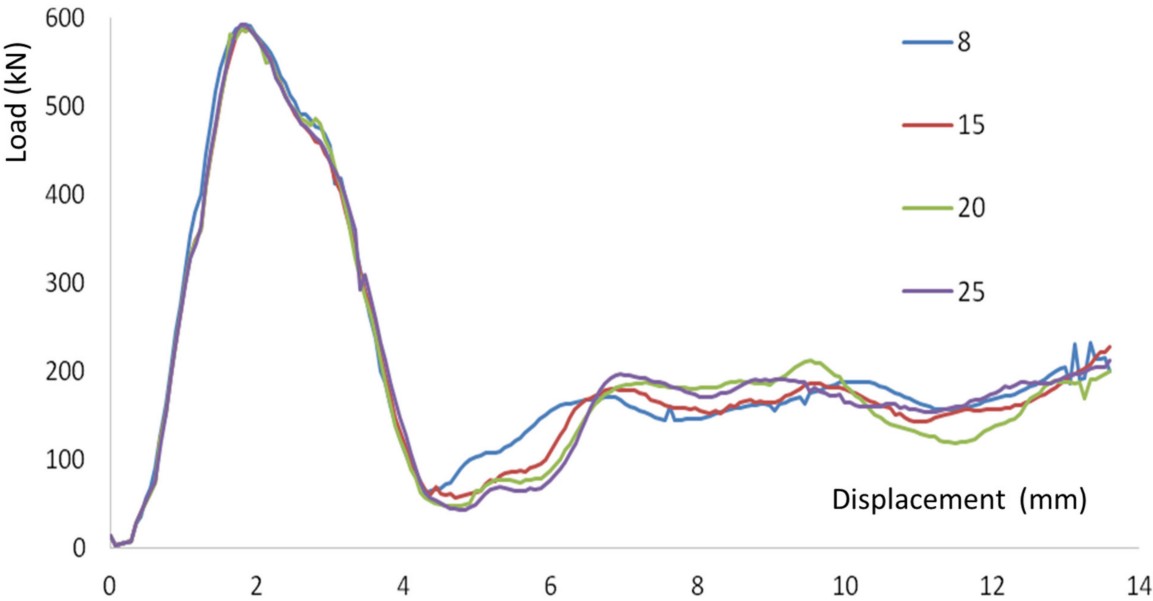

**Figure 7.** Force–displacement graphs—slab local seeding.

The optimum mass scaling value was found using a coarse mesh (Figure 8). The minimum time increment was decreased from $1 \times 10^{-5}$ to $1 \times 10^{-8}$, decreasing by a factor of 10 each time. The minimum time increment was decreased until convergence, which was achieved at $5 \times 10^{-7}$.

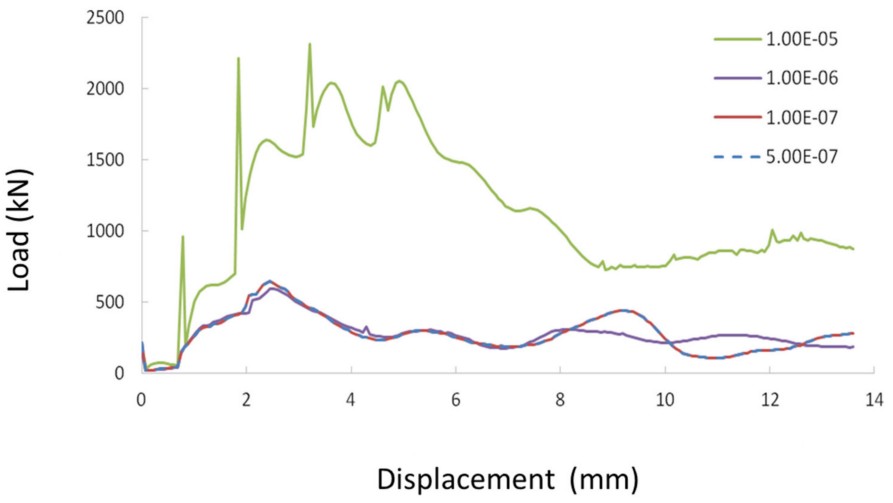

**Figure 8.** Mass Scaling Force–displacement Graphs.

The experimental force–displacement curve is shown with the force–displacement curve of the final numerical models for the column and slab members in Figures 9 and 10 below, respectively.

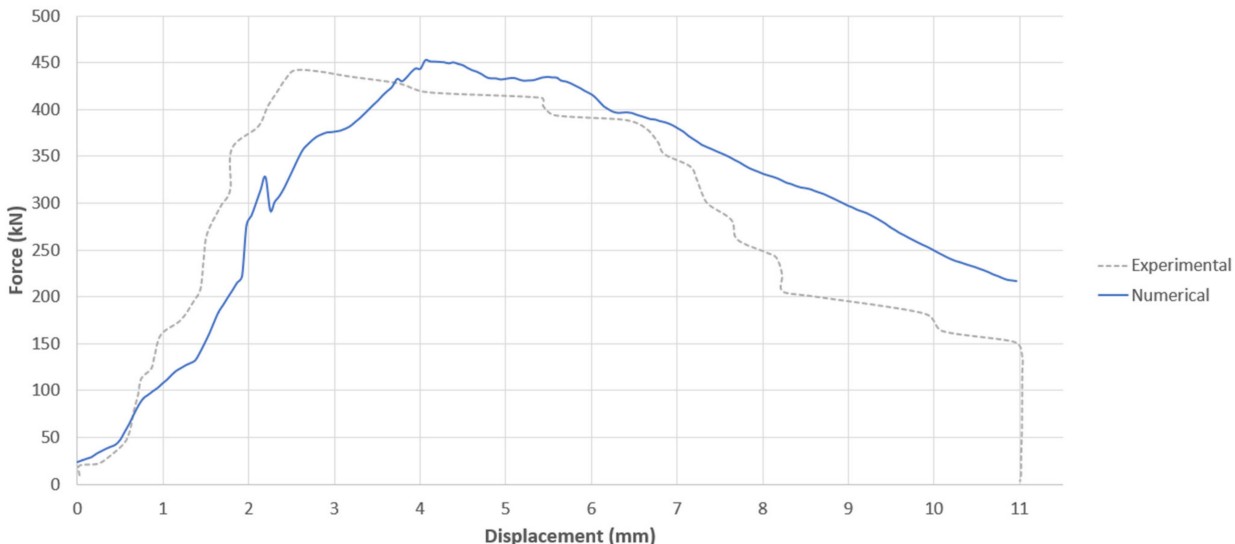

**Figure 9.** Experimental and numerical force–displacement curve for the Chen et al. studies [5].

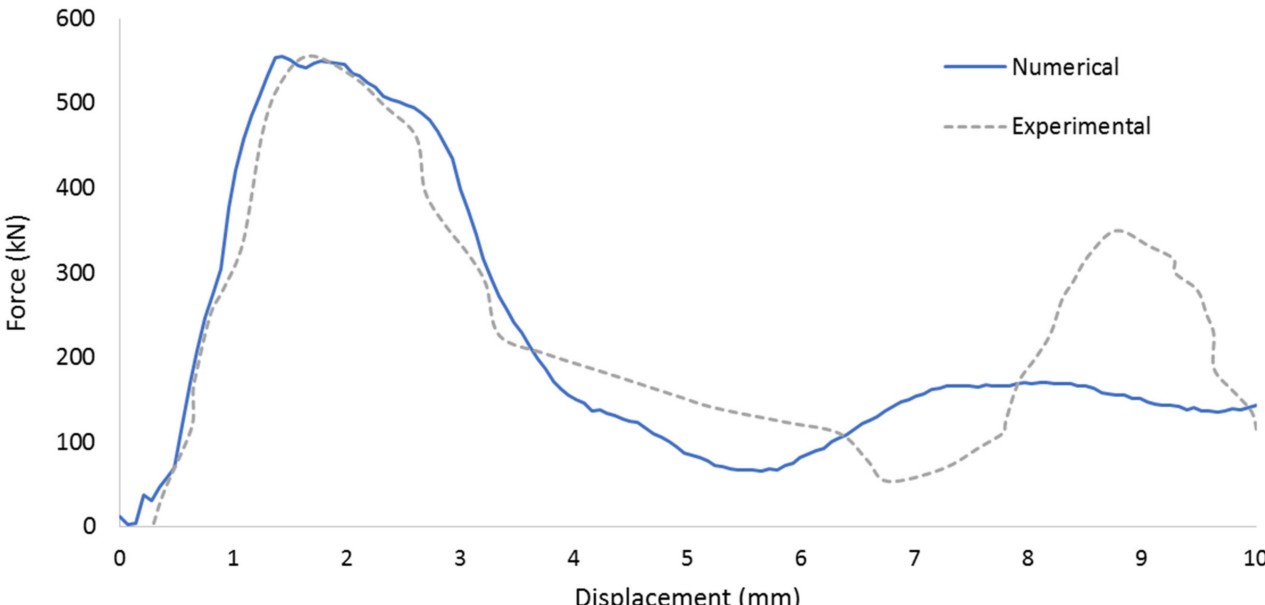

**Figure 10.** Experimental and numerical force–displacement curve for the Hrynyk and Vecchio studies [21].

### 2.4. Parametric Studies

The FRP wrapping technique is applied to the validated numerical models. The parameters which directly influence the structural response of the members subjected to impact loading is investigated. Obtaining the energy-based responses and making qualitative observations is the main purpose of the parametric studies. Hence, the dissipated energy of the section, $E_{diss}$ will be a major performance measure in this study. Quantifying how $E_{diss}$ changes will contribute to the analysis and design of FRP-strengthened CFST members for various structural applications.

The final mesh configuration of the FRP-strengthened members is summarised in Tables 10 and 11 for the CFST column and RC slab specimens, respectively.

**Table 10.** Final mesh configuration of FRP-strengthened CFST column.

|  |  | *Final Mesh Size* |
|---|---|---|
| *Steel* | *Global* | 25 |
|  | *Local* | 9 |
| *Concrete* | *Global* | 20 |
|  | *Local* | 6 |
|  | *Weight* | 11 |
|  | *Support* | 11 |
| *FRP* | *Global* | 23 |
|  | *Local* | 6 |

**Table 11.** Final mesh configuration of FRP-strengthened reinforced concrete (RC) slab.

| *Part* | *Slab* | | *Supports* | *Drop Weight* | *Steel Reinforcement* | *FRP* |
|---|---|---|---|---|---|---|
|  | *Impact Region* | *Outer Region* |  |  |  |  |
| *Seed size* | 8 | 20 | 23 | 16 | 16 | 2 |

The parameters under investigation for the CFST column are (Table 12):

- FRP: Fracture energy, thickness, bond length
- CFST: Concrete Young's Modulus, steel yield strength

**Table 12.** Values used in parametric testing.

| *Parameter* | *Unit* | *Min. Value* | *Max. Value* |
|---|---|---|---|
| *FRP fracture energy, $E_{fracture}$* | mJ/mm$^3$ | *0.10* | *250* |
| *FRP thickness, $t_{FRP}$* | mm | *0.10* | *0.50* |
| *FRP bond length, $l_{bond}$* | mm | *150* | *1500* |
| *Concrete Young's Modulus, $E_{conc}$* | GPa | *20* | *40* |
| *Steel yield strength, $f_{y, \, st}$* | MPa | *245* | *600* |

The parameters under investigation for the CFST column are (Table 13):

- FRP: Fracture energy, thickness, strip width
- Impact velocity of the drop weight, compressive strength of concrete

**Table 13.** Summary of range of tested parameters.

| Parameter | $E_{frac}$/mJ/mm$^3$ | CFRP Thickness/mm (%) | CFRP Width/mm (%) | Impact Velocity/mm/s | Concrete Compressive Strength/MPa |
|---|---|---|---|---|---|
| Fixed | 150 | 2 | 70 | 8000 | 69.4 |
| Range | 10–250 | 1.2–3.2 (0.923–2.462) | 30–110 (1.667–6.111) | 1000–12,000 | 30–69.4 |

Relative thickness and width were calculated as the dimensions of the slab were significantly smaller than the actual dimensions used in practice.

$$Relative\ thickness,\ t_r(\%) = \frac{CFRP\ thickness}{Slab\ thickness} \times 100 \tag{4}$$

$$Relative\ width,\ w_r(\%) = \frac{CFRP\ width}{Slab\ width} \times 100 \tag{5}$$

### 2.5. Results and Discussion

#### 2.5.1. CFST Column

Fracture Energy

Values of the FRP fracture energy are varied to study the influence of the amount of energy required for the failure of the FRP material.

The proposed relationship between $E_{fracture}$ and $E_{diss}$ can be well-approximated by a linear relationship between the two variables (Figure 11). This is given as:

$$y = 14.792x + 23210 \tag{6}$$

for $0.1\ \text{mJ/mm}^3 \leq E_{fracture} \leq 250\ \text{mJ/mm}^3$ where $x = E_{fracture}$, $y = E_{diss}$.

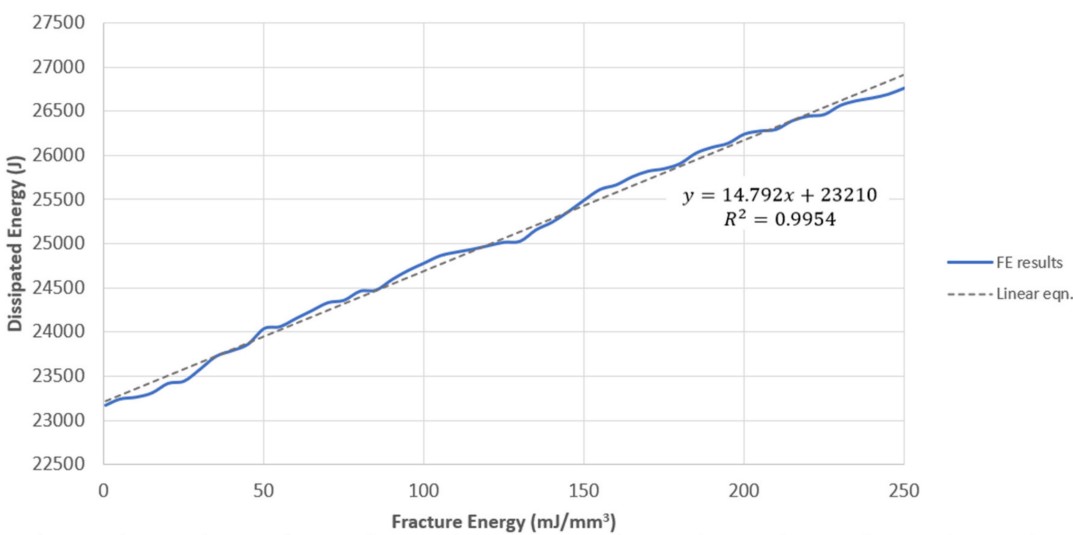

**Figure 11.** Dissipated Energy vs. Fracture Energy graph.

If an FRP with a low $E_{fracture}$ is used, FRP failure is expected to be sudden and catastrophic. Cracking and fibre damage in the FRP are anticipated to cause rapid resistance reduction. Hence, if $E_{fracture}$ is neglected in impact design, damages may be shown to be more severe.

In the case of increasing the fracture energy of the FRP material, an element with a higher $E_{diss}$ is able to withstand higher magnitudes of impact force without undergoing failure. This suggests FRP materials which require a greater energy for failure have a greater energy dissipation capacity, as shown in Figure 12 below.

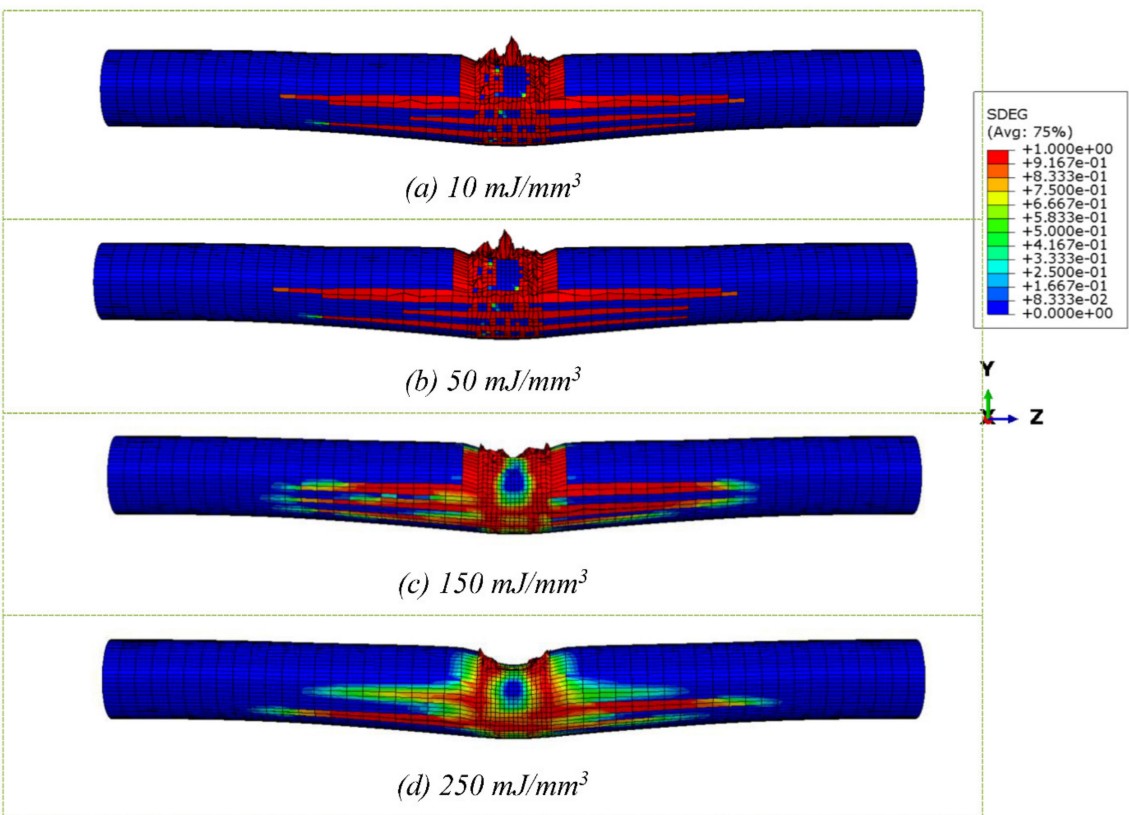

**Figure 12.** SDEG (failure criterion) for different FRP fracture energy.

Figure 12 above also suggests how the extent of immediate failure gradually decreases as an FRP with a higher $E_{fracture}$ is used. The material damage becomes less spread out from the impact zone, and its severity decreases. This enables the FRP to provide further strengthening to the column.

FRP Thickness

The relationship between $t_{FRP}$ and $E_{diss}$ may be approximated by the polynomial relationship:

$$y = -17.9 \times 10^6 x^5 + 31.8 \times 10^6 x^4 - 21.6 \times 10^6 x^3 + 6.83 \times 10^6 x^2 - 1.02 \times 10^6 x + 81.9 \times 10^3 \quad (7)$$

for 0.10 mm $\leq t_{FRP} \leq$ 0.50 mm, where $x = t_{FRP}$, $y = E_{diss}$.

The variation of $E_{diss}$ with increasing $t_{FRP}$ is shown to be nonlinear. The behaviour can be divided into three main stages. In Stage 1, $E_{diss}$ decreases by approx. 15% as $t_{FRP}$ is increased from 0.10 mm to 0.15 mm. In Stage 2, this significant rate of reducing $E_{diss}$ appears to level out near a $t_{FRP}$ value of 0.15 mm until 0.30 mm. Detailed observations of Stage 2 are given in Figure 12. In Stage 3, $E_{diss}$ continues to decrease again from 0.30 mm to the maximum tested $t_{FRP}$ value of 0.50 mm. $E_{diss}$ decreases by approx. 21% between 0.30 mm to 0.50 mm. However, the rate of decrease in $E_{diss}$ in Stage 3 is less substantial than it was initially, as this section of the curve is less steep.

Hence, Figure 13 indicates that increasing $t_{FRP}$ at lower values will have a more significant effect on lowering the $E_{diss}$ than at higher values. Additionally, there is a region where $E_{diss}$ does not seem to vary with changes in $t_{FRP}$.

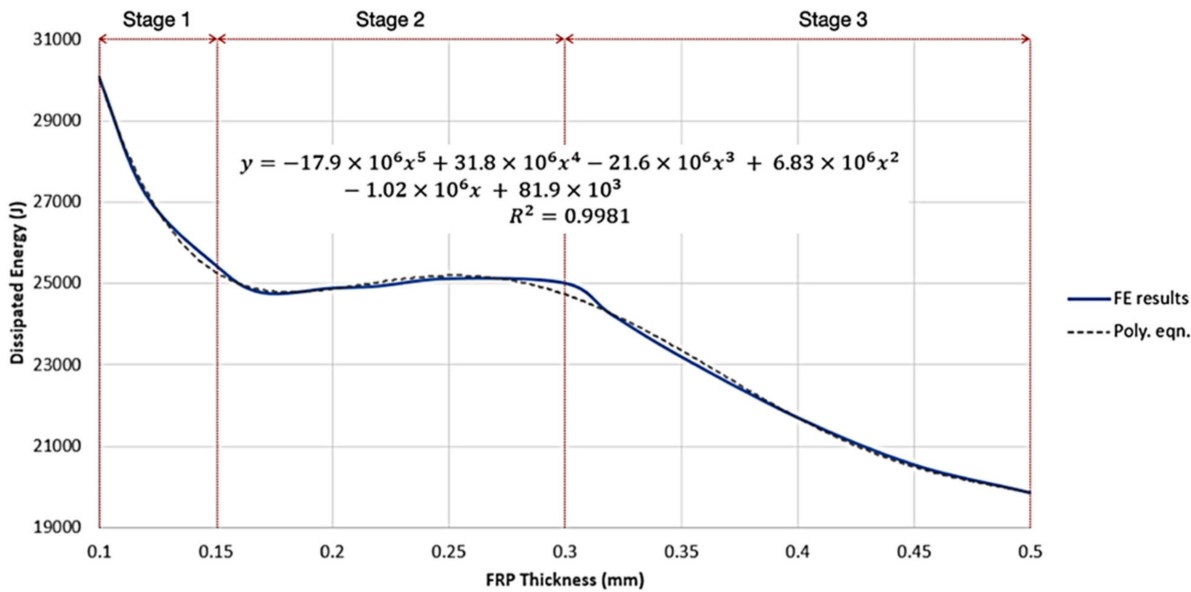

**Figure 13.** Dissipated Energy vs. FRP Thickness graph.

The overall tendency for $E_{diss}$ to decrease with increasing $t_{FRP}$ from Stage 1 to Stage 3 is observed. The force–displacement curves for $t_{FRP}$ values of 0.12 mm, 0.22 mm, and 0.40 mm are plotted together in Figure 14 below. These $t_{FRP}$ values are deemed to be representative of Stage 1, Stage 2, and Stage 3 of the $E_{diss}$ vs. $t_{FRP}$ curve, respectively. Henceforth, they are denoted as $t_{FRP,1}$, $t_{FRP,2}$, and $t_{FRP,3}$, accordingly. Figure 14 also indicates how increasing $t_{FRP}$ reduces the maximum forces reached by the impactor. Since this results in a smaller force–displacement curve overall, $E_{diss}$ is reduced as $t_{FRP}$ increases. This effect was observed was because force and displacement readings were measured at the interface between the impactor and the FRP for consistency. Hence, it is the force experienced by the drop weight that is obtained. Increasing the FRP thickness caused a reduction in the contact force by minimising the contact interaction between the member and the impact load. Thus, the maximum reaction force experienced by the impactor was reduced. This also enabled thicker layers of FRP to settle to lower minimum forces in the plateau stage. Furthermore, the force–displacement curve of $t_{FRP,1}$ shows considerably less stable behaviour. There are greater fluctuations in the force indicating significant vibrations upon impact. $t_{FRP,2}$ and $t_{FRP,3}$ show more stable responses to the impact load and display smoother curves in comparison.

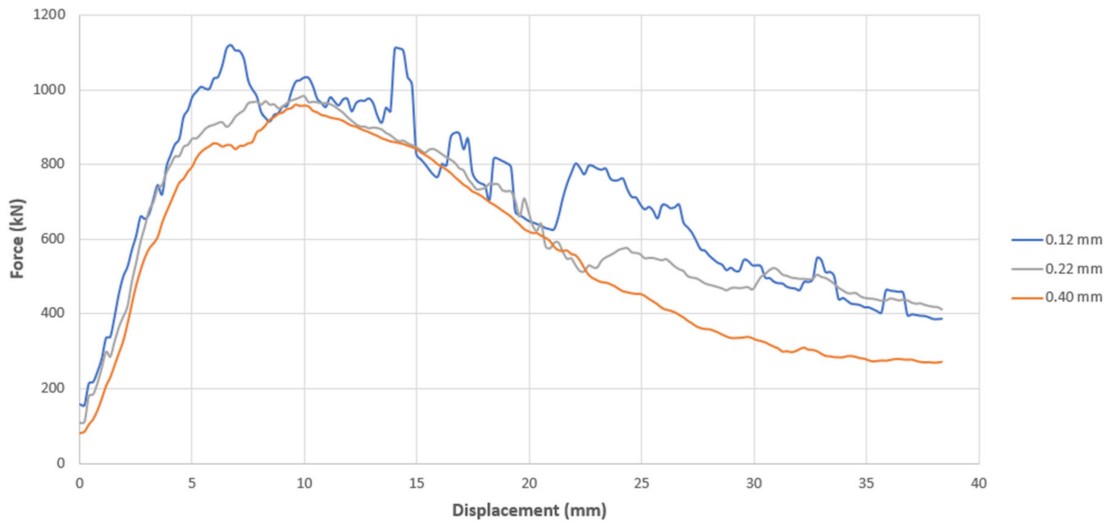

**Figure 14.** Force–displacement graph showing general behaviour of increasing FRP thickness.

The failure behaviour simulated by the FE models for the FRP wraps is given in Figure 15. This is to understand these trends in terms of the damages experienced by the FRP as $t_{FRP}$ is varied. Again, the SDEG output variable is used to show the overall FRP material damage.

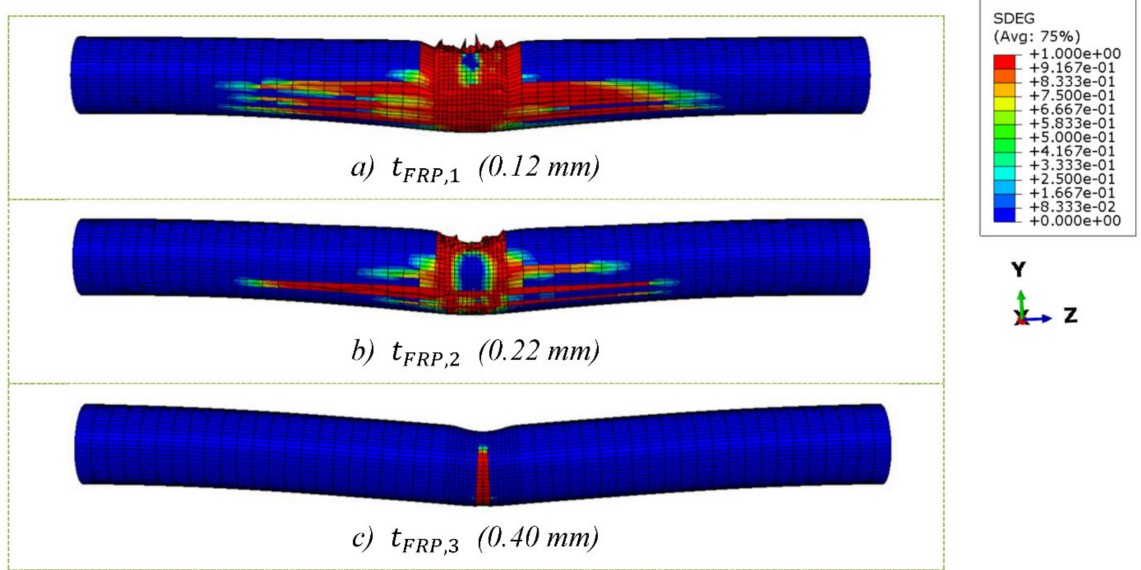

**Figure 15.** Failure behaviour with increasing different FRP thickness.

Degradation reduces as $t_{FRP}$ increases, as the FRP wrap undergoes less plastic damage. Failure propagation is less progressed at a higher $t_{FRP}$, as the longitudinal rupture of the FRP element on the underside of the column decreases in severity. For $t_{FRP,1}$, the FRP wrap has undergone catastrophic failure throughout the entire mid-span impact region at the end of the observed time period (Figure 15a). In contrast to $t_{FRP,3}$, the rupture on the underside of the FRP wrap has not progressed to the impact zone (Figure 15c).

From this performance perspective, a structural member with a lower $E_{diss}$ is favourable, as less impact energy is absorbed and the member experiences less deformation. Increasing the FRP wrap thickness results in a specimen with a lower $E_{diss}$. This indicates the member can withstand higher magnitudes of impact force without undergoing damage, hence increasing its deformability.

FRP Bond Length

The range FRP bond lengths modelled were from 150 mm $\leq l_{bond} \leq$ 1500 mm. This range is intended to represent the conditions where only the impacted mid-span is strengthened to when almost the entire 1700 mm column is strengthened by the FRP. This corresponds to $0.09 \leq \frac{l_{bond}}{L} \leq 0.90$, in terms of proportion of the column length covered by FRP wrapping. This measure of bond ratio is adopted for analysis, to allow comparison between columns of different lengths.

The equation proposed to predict the relationship $\frac{l_{bond}}{L}$ and $E_{diss}$ is given as:

$$y = -76769x^4 + 232587x^3 - 251594x^2 + 117338x + 4581.9 \tag{8}$$

for $0.09 \leq \frac{l_{bond}}{L} \leq 0.90$, where $x = \frac{l_{bond}}{L}$, $y = E_{diss}$.

Figure 16 predicts a non-linear increase in $E_{diss}$ as $\frac{l_{bond}}{L}$ increases. $E_{diss}$ increases more significantly from an $\frac{l_{bond}}{L}$ ratio of 0.09 to approx. 0.30, as the curve has a steeper gradient in this region. The curve continues to increase at a reduced rate for values of $\frac{l_{bond}}{L}$ above 0.35. A transition stage is observed between these two distinct regions of varying $E_{diss}$ increase.

This indicates that beyond this region, increasing $l_{bond}$ will not significantly change the amount of $E_{diss}$ achieved by the member.

**Figure 16.** Dissipated Energy vs. Bond Length Ratio graph.

The SDEG (failure criteria) output of the 0.20, 0.50, 0.70, and 0.80 ratio FRP at the end of the time period is compared in Figure 17. The damage propagation in the 0.20 FRP has progressed to failure. The FRP has undergone severe cracking in the underside and fibre breakage at the impact zone. The edges are no longer in contact with the steel tube. Under experimental conditions, the FRP may completely delaminate. However, a limitation of this model is its inability to convey the effects of total delaminating and adhesive failure. The 0.50 FRP is representative of the end of the transition region on the $E_{diss}$ vs. $\frac{l_{bond}}{L}$ graph and shows the damage just reaching the edge of the FRP. This implies the transition region is directly related to the extent of the FRP failure propagation. Although the 0.70 FRP also experiences significant plastic damage, cracking has not spread to the ends. The edges of the FRP are still in full contact with the steel tube and remain as one element with the CFST section. Sufficient resistance against deformation is obtained at 0.70, explaining the decreased gradient in the $E_{diss}$ vs. $\frac{l_{bond}}{L}$ graph in this region. At a 0.80 ratio, the failure mechanism remains the same for this impact scenario, as FRP damages do not spread any further.

Comparison of the FRP damages indicate how increasing the $\frac{l_{bond}}{L}$ ratio changes the damage propagation of the FRP and the column member itself. The total failure of the FRP is likely to occur at lower bond lengths. As the FRP degrades, it no longer provides strength enhancement to the CFST. This reduces the resistance of the column against impact events.

The critical ratio is dependent on the length at which total FRP failure is avoided. It should be considered for other loading conditions, including various drop hammer sizes and magnitudes. There is an increased risk of complete delaminating at a lower $\frac{l_{bond}}{L}$. Early failure due to cracking of the FRP from fibre breakage at the impact zone is more likely, as there is a shorter length of FRP for the cracks to propagate through. Contact between the FRP and steel tube will be lost more rapidly.

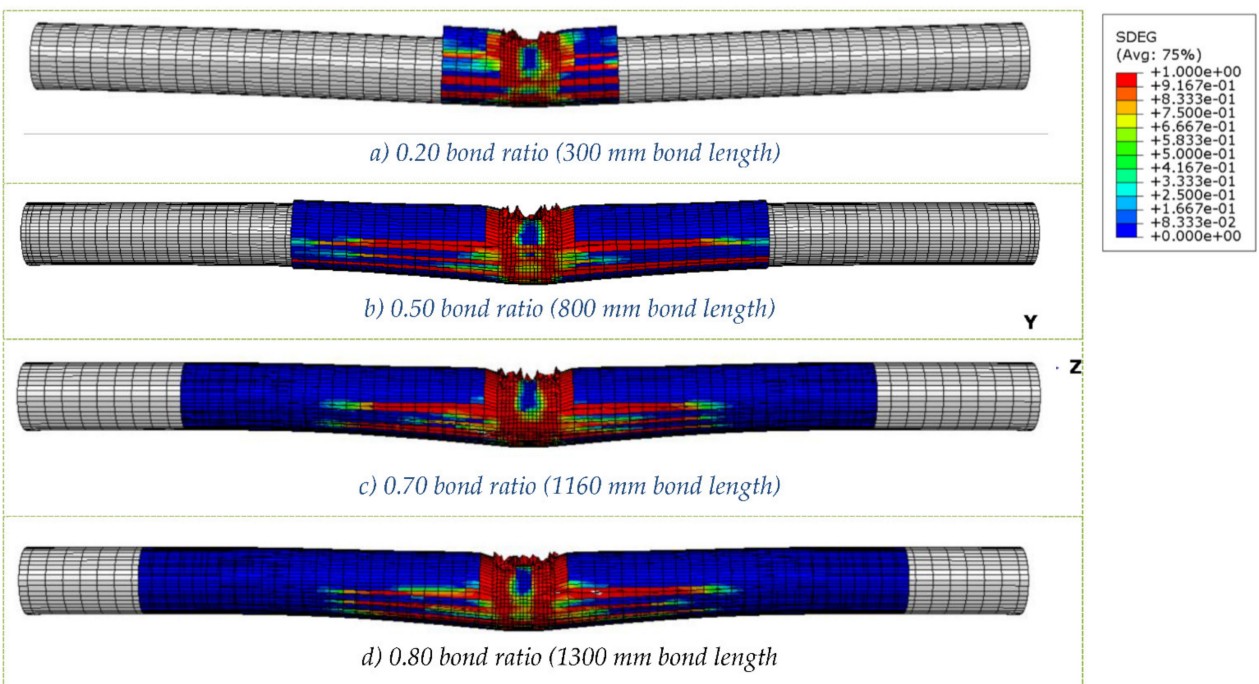

a) 0.20 bond ratio (300 mm bond length)

b) 0.50 bond ratio (800 mm bond length)

c) 0.70 bond ratio (1160 mm bond length)

d) 0.80 bond ratio (1300 mm bond length

**Figure 17.** FRP damage for different bond length ratios.

Young's Modulus of Concrete

The Young's Modulus of the concrete used in the CFST is modified between the range 20 GPa $\leq E_{conc} \leq$ 40 GPa. This is intended to cover the normal concrete strength grades available for use in CFST sections [44]. The $E_{conc}$ material property is selected based on the assumption that it is not considered to be affected by strain rate [39,45]. Note that the $E_{conc}$ are the unconfined Young's Modulus values. For each value of $E_{conc}$ trialled on the FE model, the confined properties of the concrete were calculated. Concrete compression and tension properties were updated accordingly, to allow accurate modelling of the failure behaviour (Figure 18).

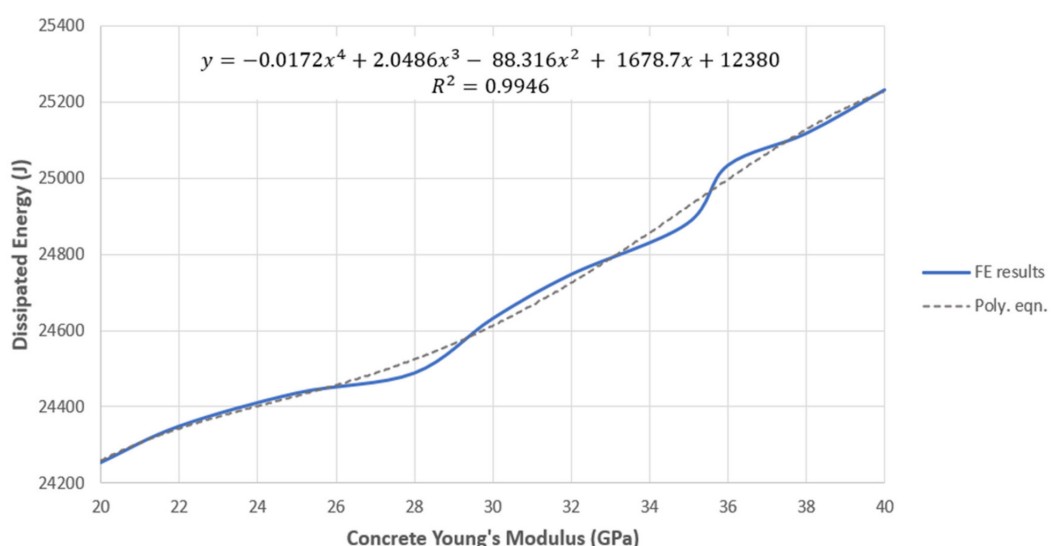

**Figure 18.** Dissipated Energy vs. Young's Modulus of Concrete graph.

The relationship between $E_{fracture}$ and $E_{conc}$ from the numerical results is suggested as:

$$y = -0.0172x^4 + 2.0486x^3 - 88.316x^2 + 1678.7x + 12380 \qquad (9)$$

for $20 \text{ GPa} \leq E_{conc} \leq 40 \text{ GPa}$, where $x = E_{conc}$, $y = E_{diss}$.

Hence, an increase in $E_{diss}$ is expected with increasing $E_{conc}$. However, the effect of increasing $E_{conc}$ is not deemed to have a significant effect on the impact response of the FRP-wrapped specimens. Only a 7% increase in $E_{diss}$ is expected as $E_{conc}$ is increased from 20 GPa to 40 GPa. This is a considerable increase in concrete grades with an insignificant performance enhancement.

Section cuts showing the maximum and minimum principal stresses developed in the concrete core are shown in Figure 19, for the two-column specimens. The concrete core exhibits high compressive stresses in the impact area. As expected, both tensile and compressive stresses developed in the 40 GPa specimen is greater.

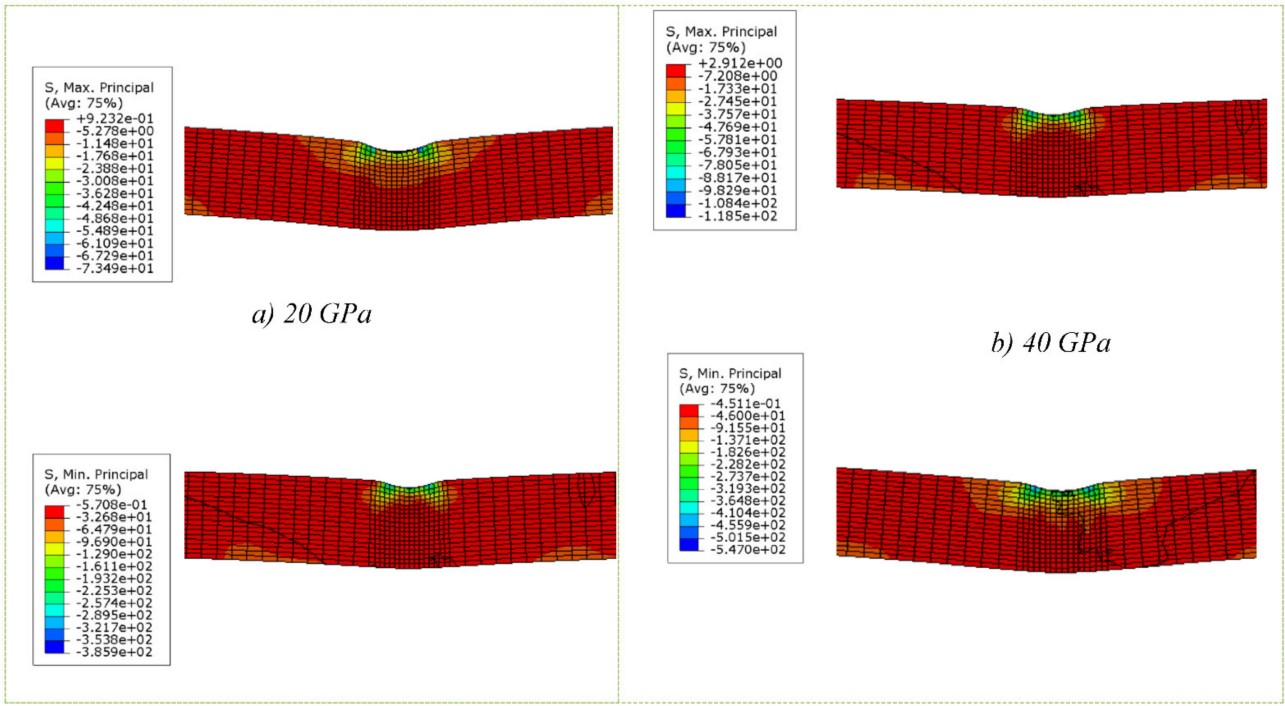

**Figure 19.** Maximum and minimum principal stress in concrete for different Young's Moduli (MPa).

The results demonstrate the minor effect of increasing concrete strength on the impact response of the columns. Although maximum compressive concrete stresses increased by 61%, this did not significantly contribute to a higher $E_{diss}$ and deformability improvement. Hence, the role of the concrete core is secondary to the steel tube in the CFST section. Using higher concrete strength grades with greater concrete stress capacity may not be the most efficient way to increase $E_{diss}$. A lower grade concrete can be used for the column without severely decreasing its performance, due to confining effects provided by the steel tube (Figure 20).

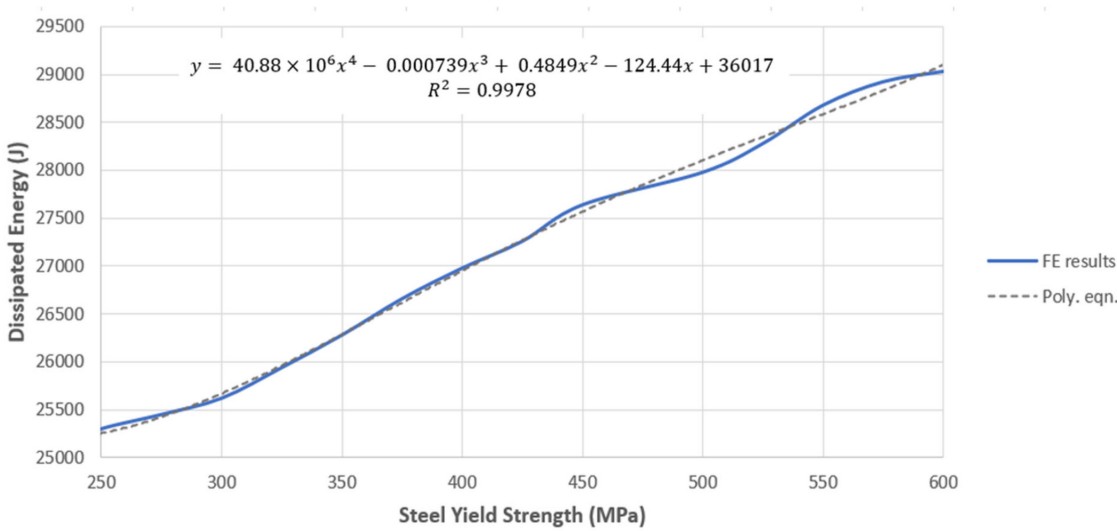

**Figure 20.** Dissipated Energy vs. Steel Yield Strength.

Steel Yield Strength

This relationship is approximated by the equation given:

$$y = 40.88 \times 10^6 x^4 - 0.000739 x^3 + 0.4849 x^2 - 124.44 x + 36017 \tag{10}$$

for 245 MPa $\leq f_{y,st} \leq$ 600 MPa, where $x = f_{y,st}$, $y = E_{diss}$.

A relatively steady increasing trend in $E_{diss}$ is predicted as $f_{y,st}$ is increased. There is an anticipated $E_{diss}$ increase of approx. 18% between this range. Hence, compared to the effect of increasing the concrete Young's Modulus, $E_c$, using a steel tube of higher $f_{y,st}$ is a more favourable way of obtaining a higher $E_{diss}$.

The PEEQ output is presented in Figure 21 for the two compared $f_{y,st}$ values. The column with the higher $f_{y,st}$ shows greater localisation of plastic deformation near the impact region. This behaviour is compared to the column with the lower $f_{y,st}$, where plastic strains are further progressed and appear more distributed. The effects of the indentation and deformations are spread more globally throughout the member, as strains are more apparent at regions away from the impact zone. Hence, a greater proportion of the tube exhibits plastic behaviour under impact loading at lower $f_{y,st}$ values.

The findings imply that a column with greater deformability and impact resistance is expected with higher $f_{y,st}$. Higher $E_{diss}$ values occur as a result of the increased column capacity. Damage due to impact loading is more localised, and a lesser proportion of the member undergoes plastic strains. Thus, the strength of the steel tube is confirmed to play a major role in increasing impact resistance (Figure 21).

2.5.2. RC Slab

Fracture Energy

The force–displacement graph for fracture energies 10, 150 and 250 mJ/mm$^3$ are shown in Figure 22.

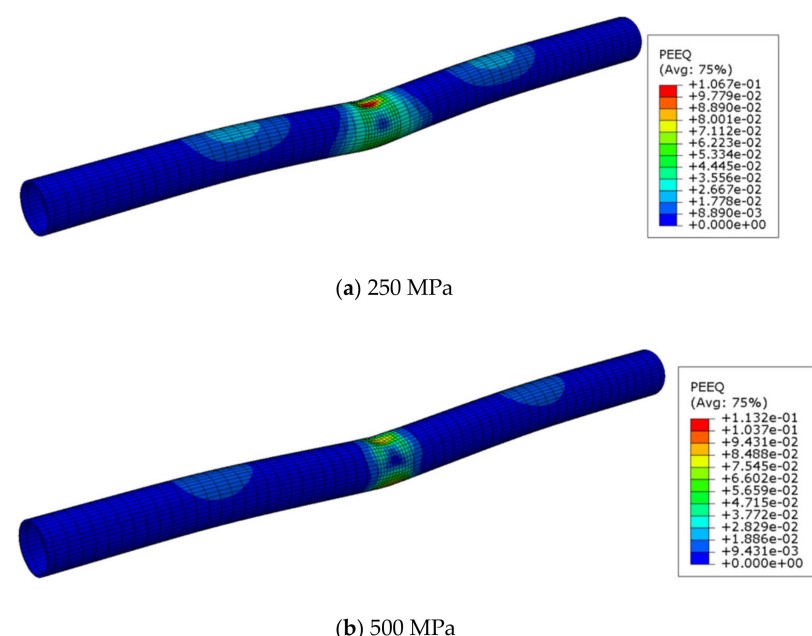

(**a**) 250 MPa

(**b**) 500 MPa

**Figure 21.** Plastic deformation in steel tube for different steel yield strengths.

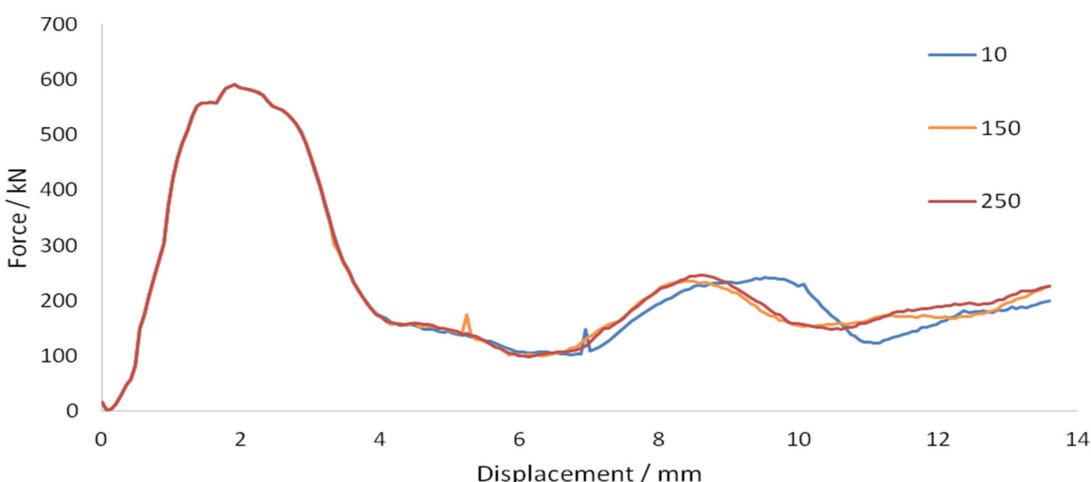

**Figure 22.** Force–displacement graphs—Fracture energy.

The graph of dissipated energy against fracture energy is shown in Figure 23.

Figure 22 shows that an increase in fracture energy of the CFRP did not influence the magnitude of the maximum force nor the shape of the first peak. However, it caused the second force peak to occur earlier at a lower displacement. An increase in fracture energy causes a slight increase in dissipated energy. This was expected as fracture energy is a ductility related parameter. An increase in fracture energy improves the ductility of the structure, which provides the same reaction force but allows the structure to deform more such that more energy is dissipated. However, the points in Figure 23 are too scattered to suggest a valid relationship between dissipated energy and fracture energy. Figure 24 shows the concrete slab crack propagation at 10 and 250 mJ/mm$^3$. Slightly less severe cracks were observed at higher fracture energy, but no significant improvement was observed.

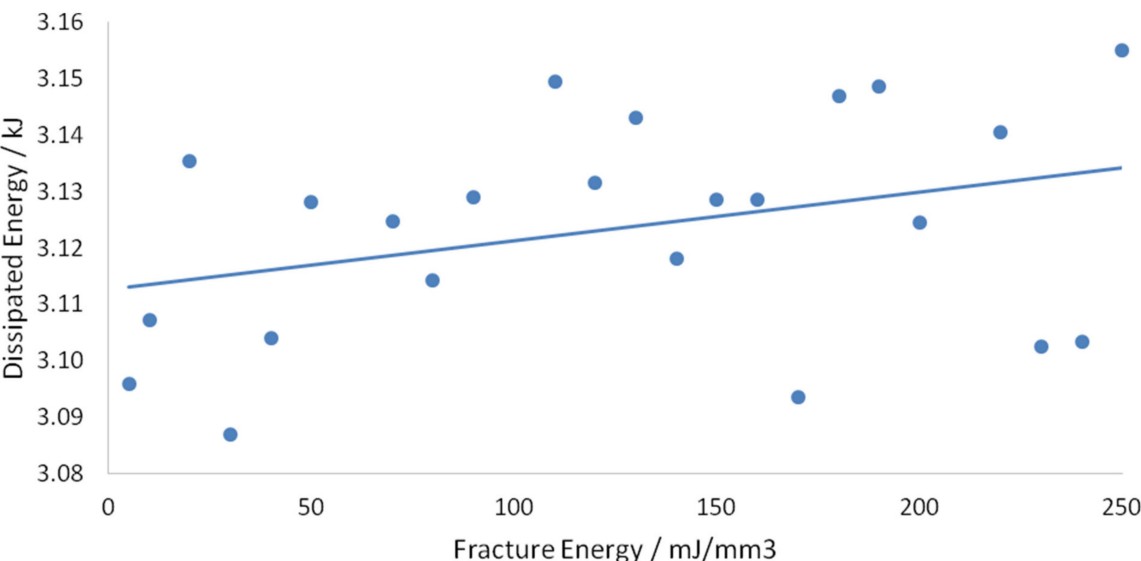

**Figure 23.** Graph of dissipated energy vs. fracture energy.

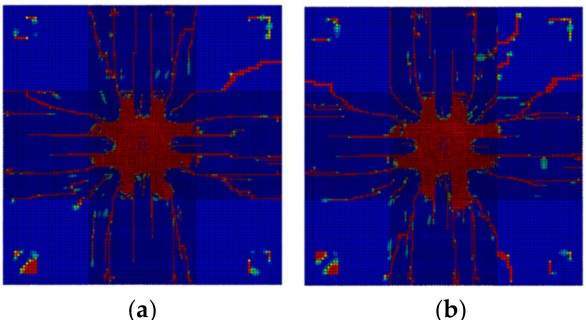

(**a**)                  (**b**)

**Figure 24.** Crack patterns at (**a**) 10 and (**b**) 250 mJ/mm$^3$ fracture energies.

CFRP Strip Thickness and Width

Figures 25 and 26 show the force–displacement graphs for 3 different thicknesses and widths, respectively.

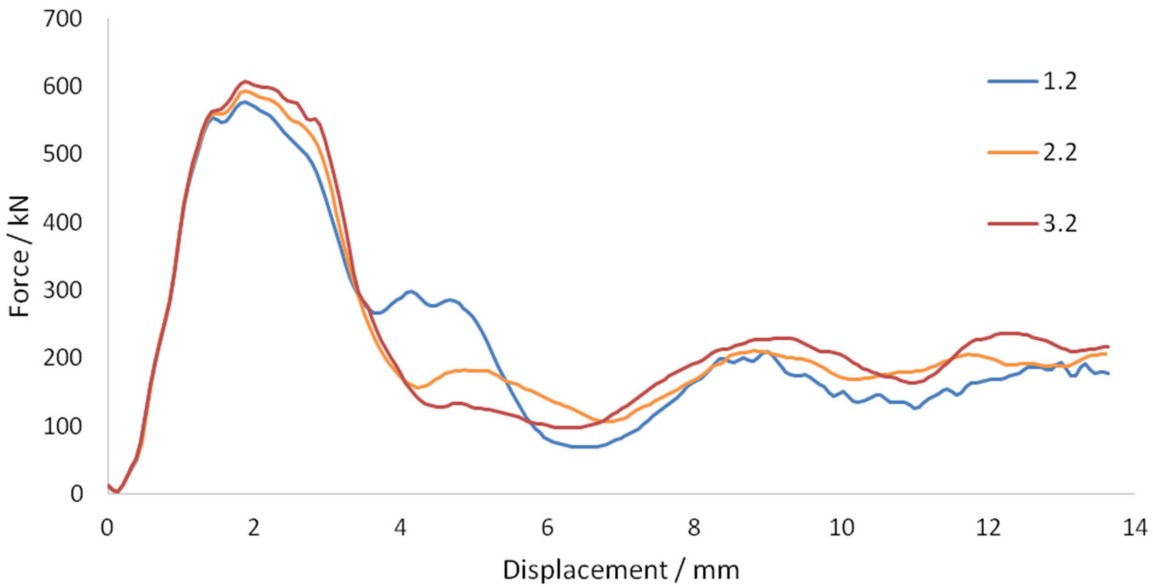

**Figure 25.** Force–displacement graphs—CFRP strip thickness.

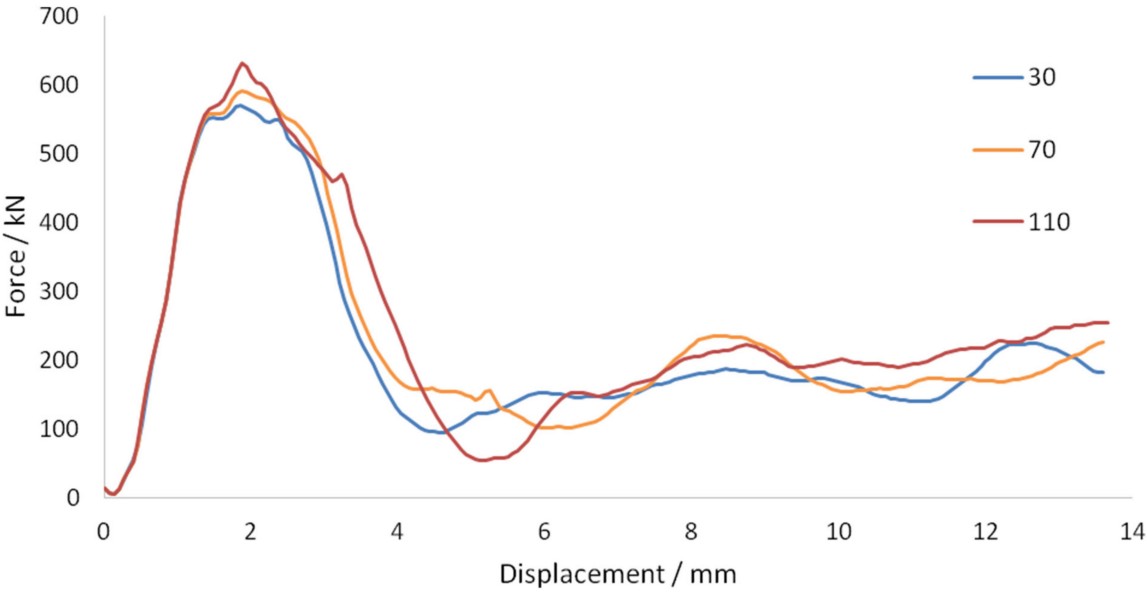

**Figure 26.** Force–displacement graphs—CFRP strip width.

The graphs of dissipated energy against relative thickness and width are shown in Figures 27 and 28, respectively.

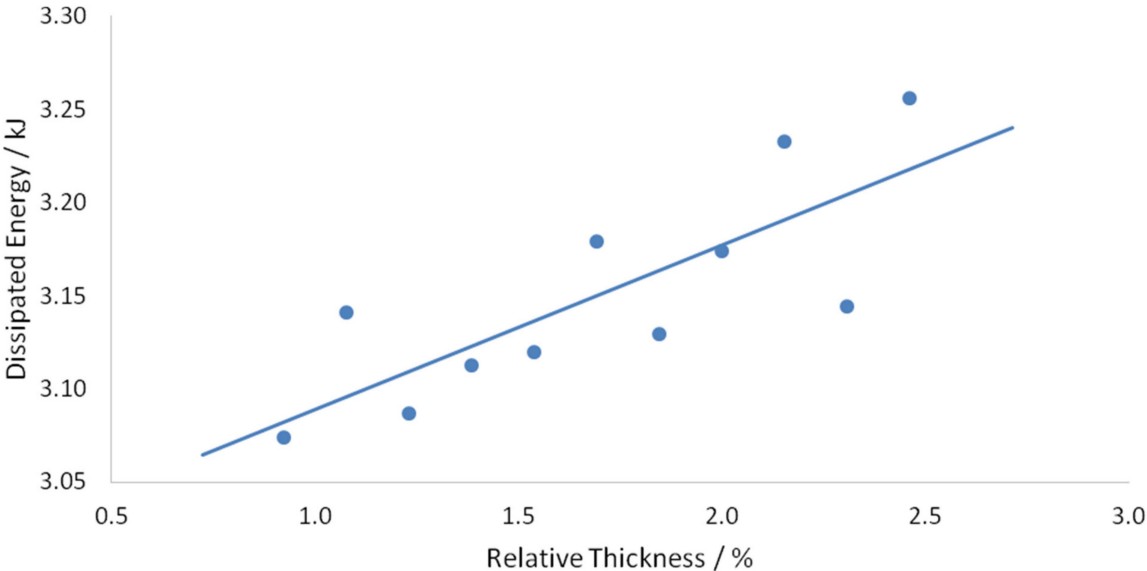

**Figure 27.** Dissipated energy vs. relative thickness graph.

Figures 25 and 26 show that greater thickness and width provided greater maximum force and greater average force after the peak force. This resulted in an increase in dissipated energy with an increase in CFRP strip dimensions, as shown in Figures 27 and 28. The CFRP strip is attached at the bottom of the slab, a distance away from the neutral axis of the cross-section. Since the CFRPs have relatively high elastic modulus, greater CFRP strip dimensions improve the initial stiffness of the structure and provide the greater maximum force. At greater strip dimensions, a more deformable material is provided, which would have led to a greater average force after the peak force.

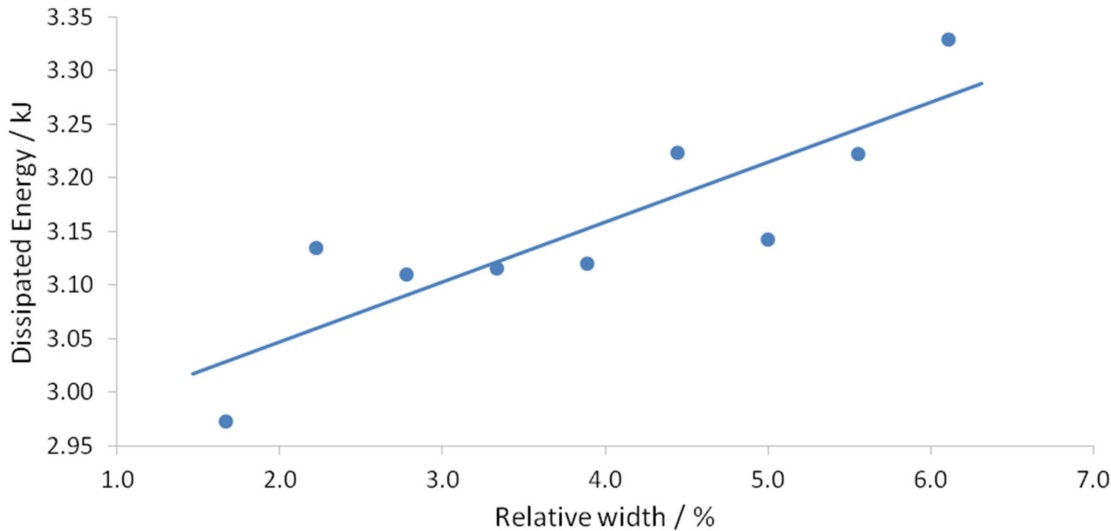

**Figure 28.** Dissipated energy vs. relative width graph.

For a given thickness and width, the total volume of the CFRP strips was calculated to compare the dissipated energy efficiency with respect to the increase in CFRP volume. The graphs of dissipated energy against CFRP volume for width and thickness are shown in Figure 29.

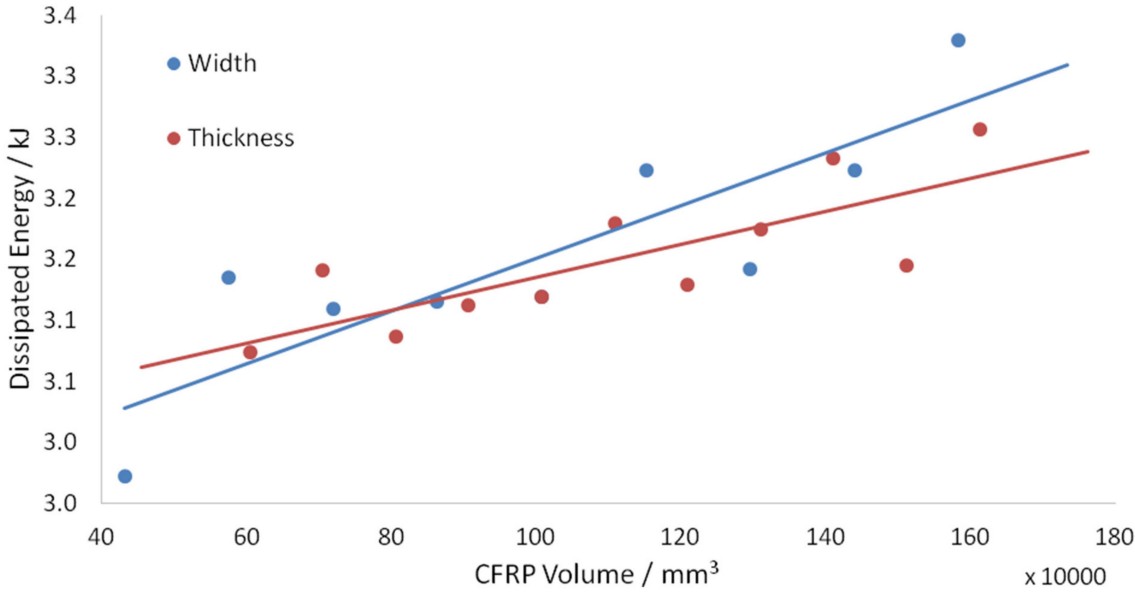

**Figure 29.** Dissipated energy against CFRP volume.

The gradient of the graphs in Figure 29 is equivalent to the increase in dissipated energy per unit increase in CFRP volume, which is indicative of the efficiency. The graph for CFRP strip width has a steeper gradient, suggesting that increasing the width is a more efficient method in obtaining greater dissipated energies. Maximum forces shown in Figures 25 and 26 for thickness and width are approximately the same. However, a less significant decrease in force from the peak force is observed for an increase in width only. Since the displacement is linearly proportional to the time in this analysis, a greater duration of maximum impulse is also suggested. The area of contact between the CFRP strip and the slab is independent of the strip thickness and dependent on the strip width. Figures 30–33 compare the stress distribution in concrete at small and large thicknesses and widths, respectively. It can be observed from these figures that an increase in width

provided a significant reduction in stresses on the underside of the slab compared to an increase in thickness.

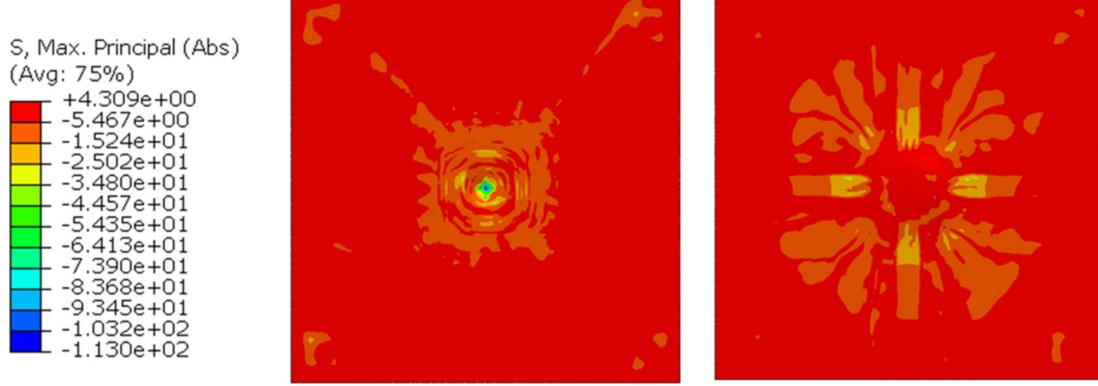

**Figure 30.** Maximum principal stress contour—1.2 mm thickness.

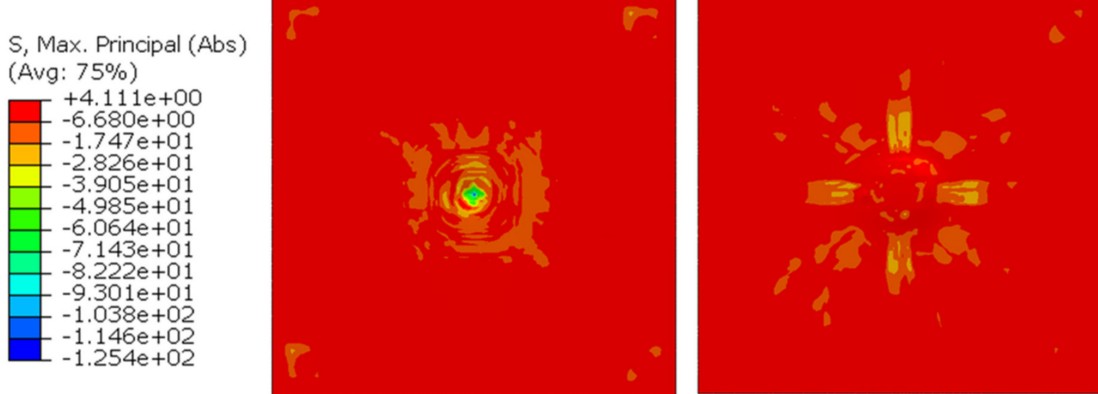

**Figure 31.** Maximum principal stress contour—3.2 mm thickness.

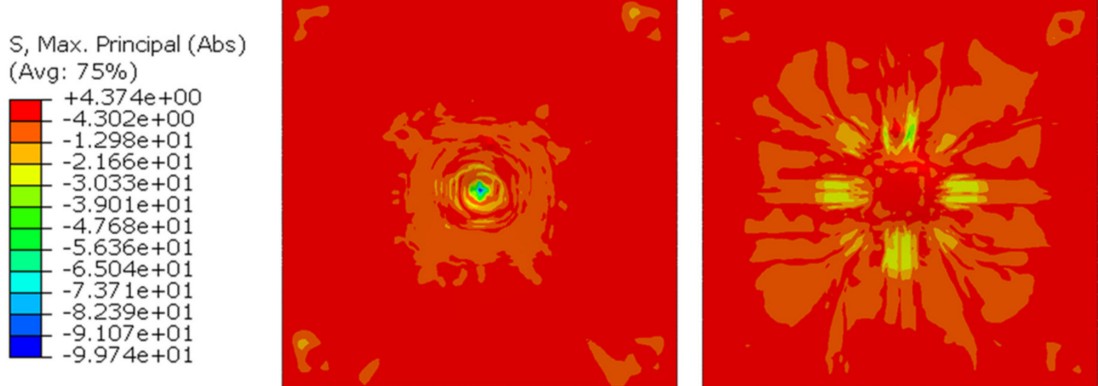

**Figure 32.** Maximum principal stress contour—30 mm width.

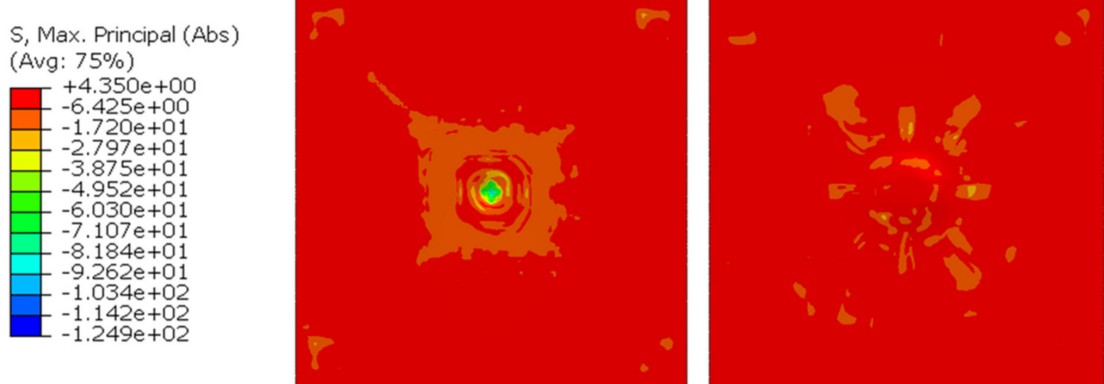

**Figure 33.** Maximum principal stress contour—110 mm width.

These results suggest that an increase in contact area may aid in distributing the forces more evenly across the structure. Therefore, resulting in a greater increase in dissipated energy per unit increase in CFRP volume via an increase in width.

Finally, the following equations are proposed for the relationship between the dissipated energy and relative thickness and width, respectively.

$$E_{diss}(t_r) = 0.0884t_r + 3.0005 \tag{11}$$

$$E_{diss}(w_r) = 0.056w_r + 2.9346 \tag{12}$$

where $t_r$, $w_r$ and $E_{diss}$ are in % and kJ, respectively.

The linear regression the $R^2$ value was 0.631 and 0.740, respectively, which may be questionable. Therefore, obtaining more data points is recommended to achieve reasonable validity.

Impact Velocity

Figure 34 compares the force–displacement graphs for the reinforced slabs with and without CFRP strips under impact velocities of 4, 7 and 10 m/s.

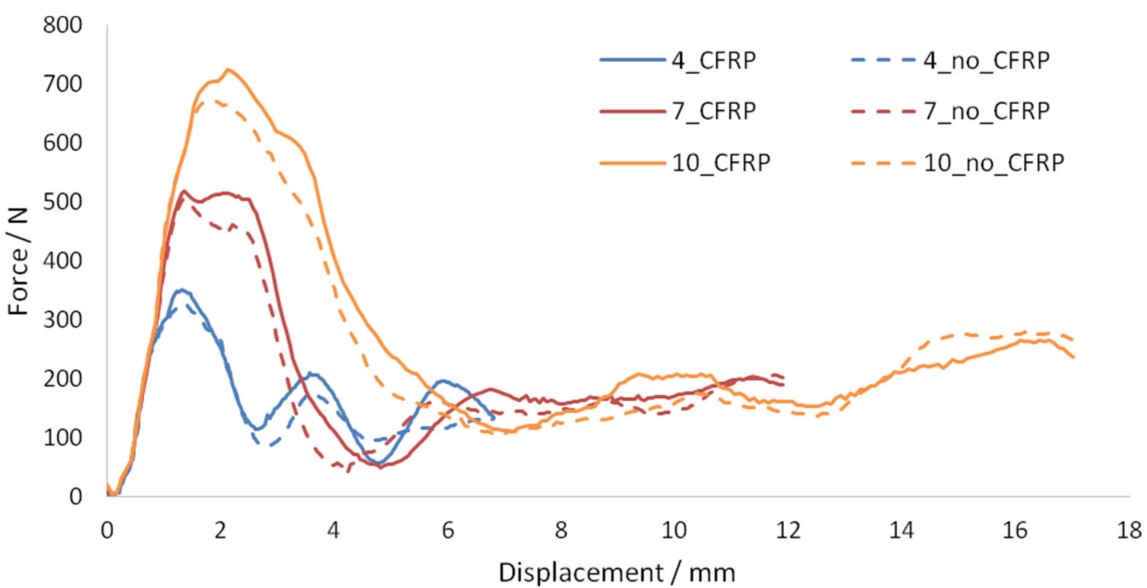

**Figure 34.** Force–displacement graphs—Impact velocity.

Figure 35 compares the graphs of dissipated energy against impact velocity for the reinforced slabs with and without CFRP.

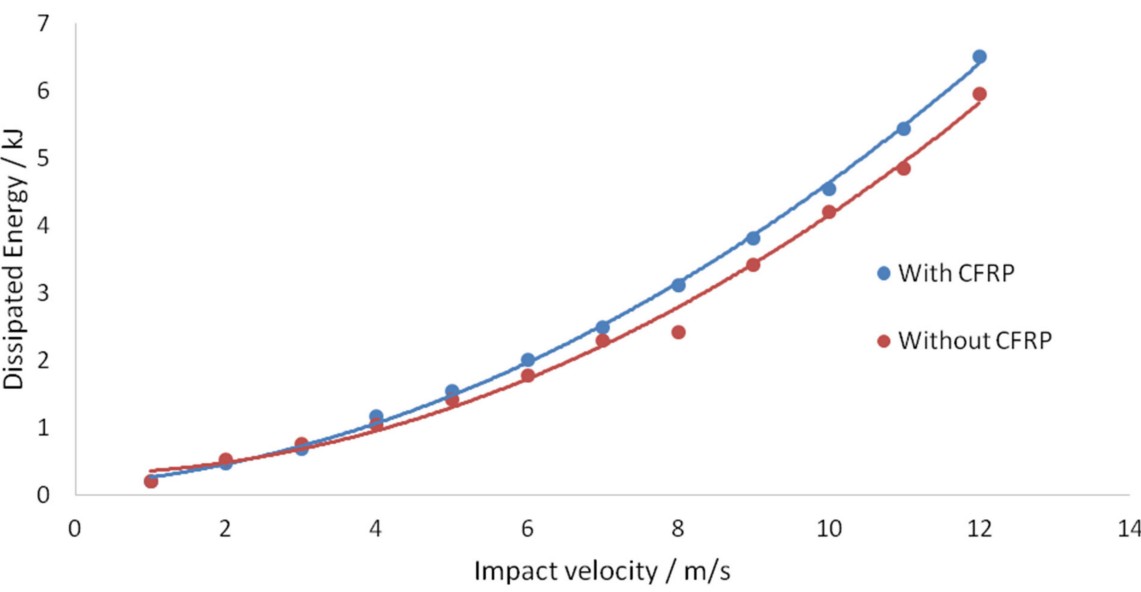

**Figure 35.** Dissipated energy against impact velocity.

As shown by Figure 34, higher impact velocities resulted in higher maximum force and displacement for the same analysis time. This led to an exponential increase in dissipated energy with increasing impact velocity, as shown in Figure 35. In both figures, the graph for the reinforced slab without CFRP is always beneath that of the slab with CFRP. This suggests an improvement in structural performance by the CFRP strips. The additional dissipated energy provided by the CFRP strips was observed to be negligible at low impact velocity and became more significant at higher impact velocities. Hence, the use of CFRP is less recommended for structures subjected to low-velocity impacts. This is also supported by Figures 36 and 37, where a significant reduction in the stresses in the slab was observed in the presence of the CFRP strips.

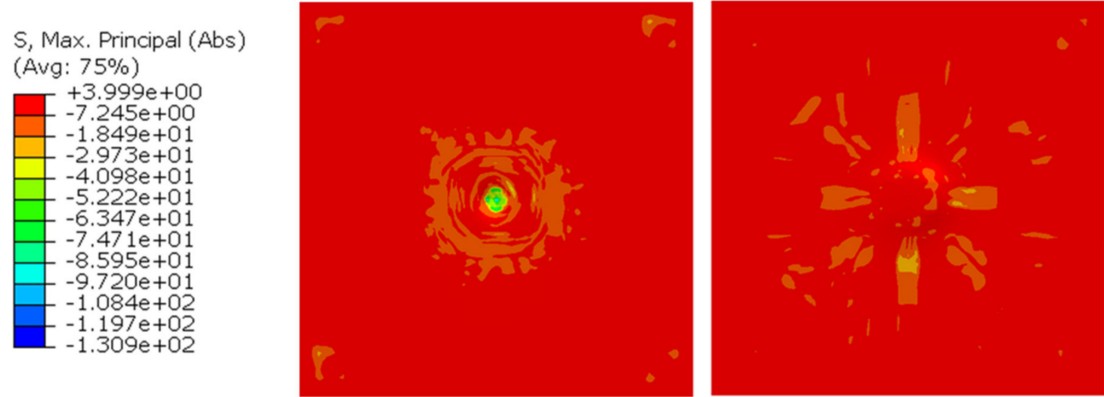

**Figure 36.** Maximum principal stress contour—12 m/s impact velocity, with CFRP.

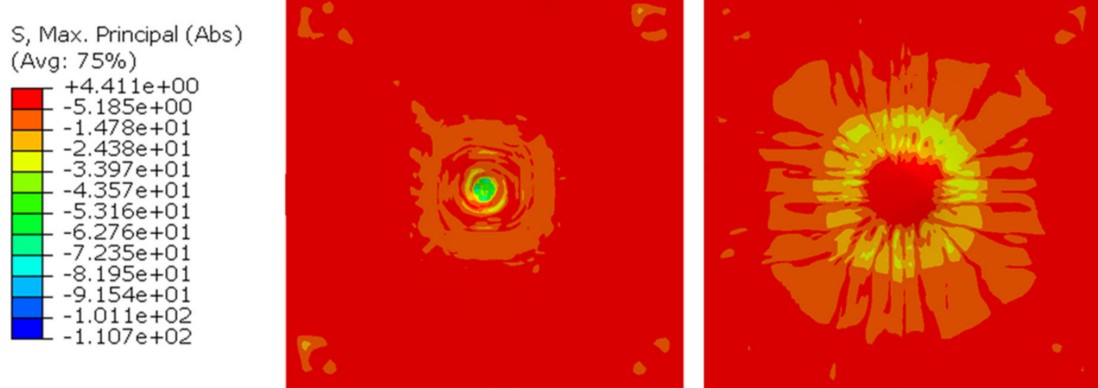

**Figure 37.** Maximum principal stress contour—12 m/s impact velocity, without CFRP.

The following equations are proposed for the relationship between dissipated energy and impact velocities for the reinforced slab with and without the CFRP strips, respectively.

$$E_{diss}(V_{CFRP}) = 0.0364V_{CFRP}^2 + 0.086V_{CFRP} + 0.1381 \tag{13}$$

$$E_{diss}(V_{no\ CFRP}) = 0.0372V_{no\ CFRP}^2 + 0.0132V_{no\ CFRP} + 0.3051 \tag{14}$$

where $v$ and $E_{diss}$ are in m/s and kJ, respectively.

The linear regression $R^2$ values were 0.999 and 0.994, respectively. Thus, the equations are accurate for the obtained data set.

Note that the model could not perform the analysis for impact velocities greater than 12 m/s due to excessive deformations. Moreover, the results at high impact velocities may be questionable as strain rate effects are more significant at these velocities.

Concrete Compressive Strength

Figure 38 compares the force–displacement graphs for the reinforced slabs with and without CFRP strips for concrete compressive strengths of 40, 50 and 60 MPa.

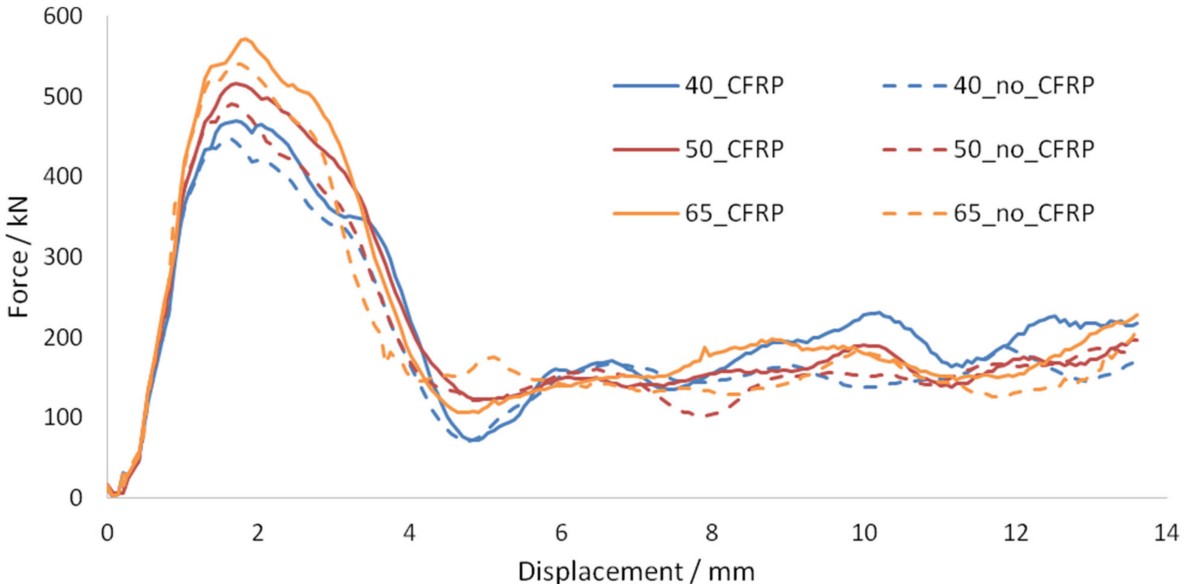

**Figure 38.** Force–displacement graphs—Concrete compressive strength.

Figure 39 compares the graphs of dissipated energy against concrete compressive strengths for the reinforced slabs with and without CFRP.

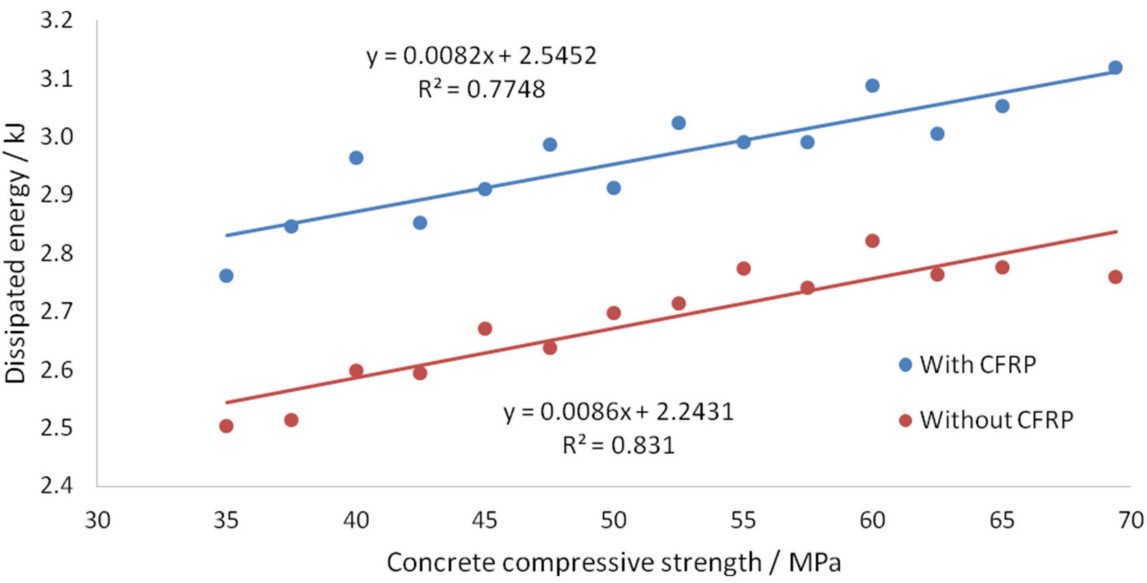

**Figure 39.** Dissipated energy against concrete compressive strength.

Similarly to impact velocity, the force–displacement trace for the reinforced slab without the CFRP always remained below that of with CFRP. A linear relationship between the dissipated energy and concrete compressive strength was observed from Figure 39. The additional dissipated energy provided by the CFRP strips remained constant throughout the range of concrete strengths tested out. Hence, CFRP strengthening will always provide similar additional dissipated energy regardless of the concrete compressive strength. This is supported by Figures 40 and 41 where the reduction in stress in concrete by the CFRP strip is similar for concrete compressive strengths of 40 and 65 MPa (Figures 42 and 43).

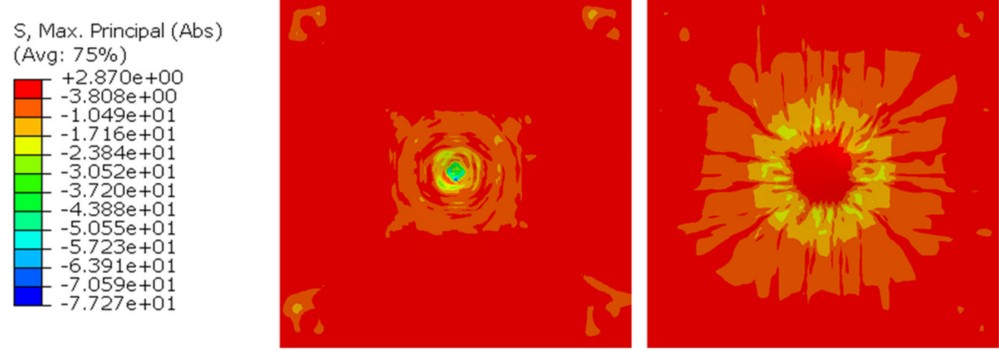

**Figure 40.** Maximum principal stress contour—Concrete compressive strength 40 MPa, without CFRP.

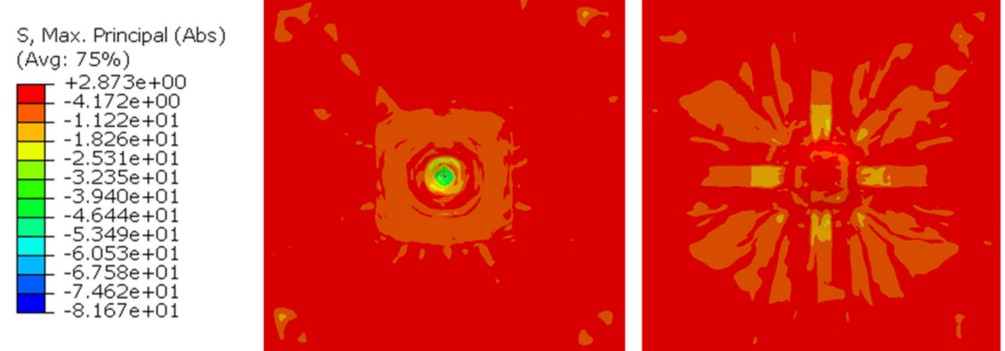

**Figure 41.** Maximum principal stress contour—Concrete compressive strength 40 MPa, with CFRP.

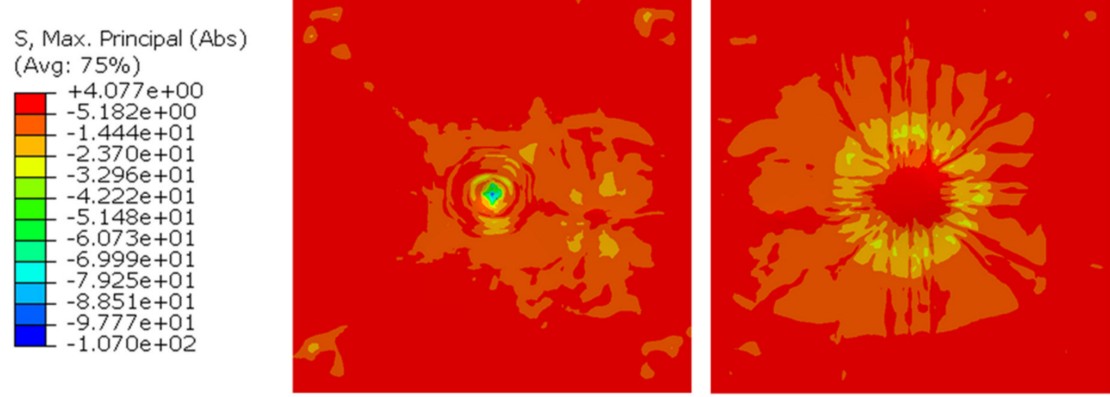

**Figure 42.** Maximum principal stress contour—Concrete compressive strength 65 MPa, without CFRP.

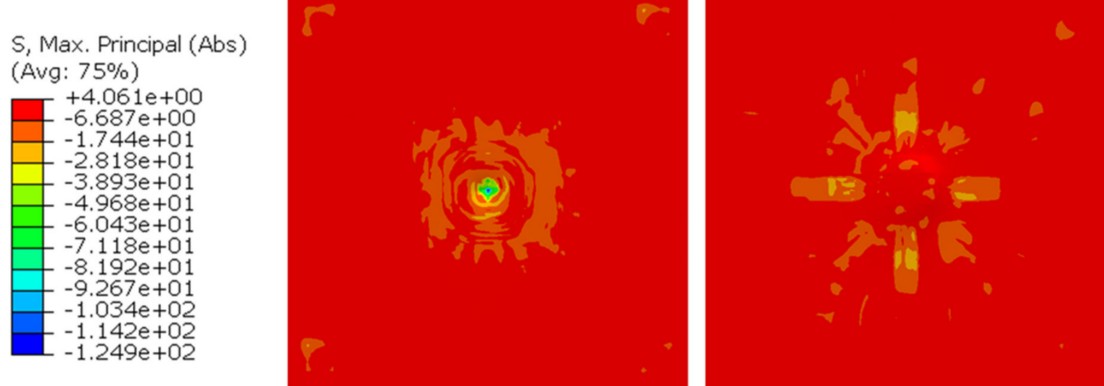

**Figure 43.** Maximum principal stress contour—Concrete compressive strength 65 MPa, with CFRP.

The following equations are proposed for the relationship between dissipated energy and concrete compressive strength for the reinforced slab with and without the CFRP strips, respectively.

$$E_{diss}\left(f'_{c,\ CFRP}\right) = 0.0082 f'_{c,CFRP} + 2.5452 \tag{15}$$

$$E_{diss}\left(f'_{c,\ no\ CFRP}\right) = 0.0086 f'_{c,\ no\ CFRP} + 2.2431 \tag{16}$$

where $f'_c$ and $E_{diss}$ are in MPa and kJ, respectively.

The linear regression $R^2$ values were 0.775 and 0.831, respectively. Thus, the equations are reasonably accurate for the obtained data set.

## 3. Conclusions

A novel method of using Abaqus/Explicit to develop FE models of a CFST column and RC slab under a dynamic impact loading condition was proposed. An agreement was found between the numerical and experimental behaviour. This confirmed the geometrical, material, and contact properties were appropriately modelled. The numerical models could closely match the force–displacement behaviour and achieve the same maximum impact force and displacement values as the experimental results. Hence, calibration of the models is a crucial task before conducting parametric studies.

Parametric studies were undertaken on a set of representative specimens, based on the validated models. The effect of various parameters on the deformability of FRP-strengthened specimens was observed. The findings and methods used in this study may be immediately applied to predict and improve the energy dissipation ability and deformability of FRP-strengthened CFST columns and RC slabs under impact loading. Since there are no existing design codes, this will aid the engineering practice when making efficient decisions when designing ductile structural members. Appropriate FRP and material prop-

erties may be selected to maximise efficiency. The developed Finite Element Method could considerably establish a good agreement with the reported experimental investigation.

**Author Contributions:** Conceptualization: F.T., L.Z. and S.P.; Literature review: All authors; Research Method: All authors; Software: L.Z., S.P. Drafting the article based on the numerical reports: F.T., S.S., Resources and interpretation: F.T., S.S. All authors have read and agreed to the published version of the manuscript.

**Funding:** This research received no external funding.

**Acknowledgments:** Authors would like to express the deepest appreciation to the University of New South Wales and the University of Sydney to provide convenient places and resources to undertake the current research.

**Conflicts of Interest:** The authors declare no conflict of interest.

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
