# Peer review of "Numerically Evaluation of FRP-Strengthened Members under Dynamic Impact Loading"

_buildings, doi:10.3390/buildings11010014_

Round 1

Reviewer 1 Report

The authors present a technical work dealing with the dynamic behaviour of FRP-strengthened reinforced concrete structures subject to impact loading. The paper is well-written and organised and its main purposes are clearly highlighted in the Introduction. The key aspect of the work resides in an accurate calibration of a Finite Element (FE) model to closely match experimental results. The model is afterwards used to perform parametric analyses and give analytical expressions useful in the design of reinforced concrete structures.

The paper is of technical interest because gives insights on the dynamic behaviour of FRP-strengthened reinforced concrete structures and furnishes details on the numerical modelling strategy to be readily extended to other structural typologies.

My specific comments are listed below.

  • It would be interesting to compare the numerical results obtained with the FE model with experimental results not used in its calibration. If such a comparison had been already performed, it should be clearly highlighted.
  • The concrete compression and tension behaviour should be given in detail, as well as the parameters of the Concrete Damage Plasticity model adopted.
  • The main findings should be highlighted in the concluding section which is rather general.
  • The authors could discuss on the possibility of extending the proposed numerical model to other structures.

Minor comments and typos:

  • Please, double-check references to Figures and Tables (see for instance line 493).

Author Response

Reviewer’s comments

Authors’ reply

The authors present a technical work dealing with the dynamic behaviour of FRP-strengthened reinforced concrete structures subject to impact loading. The paper is well-written and organised and its main purposes are clearly highlighted in the Introduction. The key aspect of the work resides in an accurate calibration of a Finite Element (FE) model to closely match experimental results. The model is afterwards used to perform parametric analyses and give analytical expressions useful in the design of reinforced concrete structures.

The paper is of technical interest because gives insights on the dynamic behaviour of FRP-strengthened reinforced concrete structures and furnishes details on the numerical modelling strategy to be readily extended to other structural typologies.

We would like to appreciate your comments.

It would be interesting to compare the numerical results obtained with the FE model with experimental results not used in its calibration. If such a comparison had been already performed, it should be clearly highlighted.

This is a very interesting suggestion; however, there are limited reported experiments in field of the impact engineering where they could provide all input data as well as the details of the experimental set up. That is the one of the main reasons that we just used the available reported experimental investigation.   

The concrete compression and tension behaviour should be given in detail, as well as the parameters of the Concrete Damage Plasticity model adopted.

Some of the references in, which were published by the authors, are added to the current text. In the presented publications further details about the concrete damage plasticity were provided. 

The main findings should be highlighted in the concluding section which is rather general.

Thanks to the comments, the relevant text is added in the conclusion part.

The authors could discuss on the possibility of extending the proposed numerical model to other structures.

Thanks to the comments, the relevant text is added in the conclusion part.

Minor comments and typos:

Please, double-check references to Figures and Tables (see for instance line 493).

Thanks to the comments, the relevant text is added in the conclusion part.

Reviewer 2 Report

REVIEWER’s REPORT

On the paper buildings-1050414

Numerically Evaluation of FRP-Strengthened Members under Dynamic Impact Loading

In this paper the author presents numerical models for adequately replicate the impact behaviour and damage process of fibre-reinforced polymer (FRP)-strengthened concrete-filled steel tube (CFST) columns and Reinforced Concrete slabs.

The results are new, correct and detailed. The methods are well described. The discussion and conclusions are adequately supported by the data. The writing is acceptable.

The paper doesn’t require a revision.

            Taking the above into consideration, I recommend the paper for publication in the journal Buildings (ISSN 2075-5309).

10.12.2020                                                            

Author Response

Reviewer’s comments

Authors’ reply

Comments from the second Authors

In this paper the author presents numerical models for adequately replicate the impact behaviour and damage process of fibre-reinforced polymer (FRP)-strengthened concrete-filled steel tube (CFST) columns and Reinforced Concrete slabs.

The results are new, correct and detailed. The methods are well described. The discussion and conclusions are adequately supported by the data. The writing is acceptable.

The paper doesn’t require a revision.

            Taking the above into consideration, I recommend the paper for publication in the journal Buildings (ISSN 2075-5309).

We would like to appreciate your comments.

Reviewer 3 Report

Dear Authors,

The topic looks fine and the manuscript organized well however lack of fundamental discussion is clear. It's highly recommended to follow the items which then helps to improve the paper.

  1. No sign of outputs/results in abstract, please revise.
  2. The introduction needs improvement. More than 80% of references are older than 2015 while such a scientific paper needs to update and have more recent similar research on the relevant topics.
  3. Are the authors have full access to the details on the references which they used as validation experiments?
  4. Fig. 2 is not positioned right, mixed inside the text, please revise.
  5. Line 249, 394, 404, 423, 449, 506, 493, 536: something missed.
  6. The results presentation looks fine however a lack of deep discussion is clear. For such a scientific paper, a precise explanation of findings is necessary.
  7. The conclusion has no sign of outputs. The results should be summarized and explained somehow in a quantified view. It should totally revise.
  8. The whole manuscript converted to PDF in the wrong way. lots of paragraphs moved, some links removed and some figures mixed. Editorial revision is nessasary.

Author Response

Reviewer’s comments

Authors’ reply

The topic looks fine and the manuscript organized well however lack of fundamental discussion is clear. It's highly recommended to follow the items which then help to improve the paper.

We would appreciate your professional comments.

No sign of outputs/results in abstract, please revise.

Thanks to the comments, the relevant text is added in the conclusion part.

The introduction needs improvement. More than 80% of references are older than 2015 while such a scientific paper needs to update and have more recent similar research on the relevant topics.

Thanks to the comments, some of the recent references were added. 

Are the authors have full access to the details on the references which they used as validation experiments?

Yes, we had access to the presented references.

Fig. 2 is not positioned right, mixed inside the text, please revise.

Thanks to the comments, it was addressed.

Line 249, 394, 404, 423, 449, 506, 493, 536: something missed.

Thanks to the comments, it was addressed.

The results presentation looks fine however a lack of deep discussion is clear. For such a scientific paper, a precise explanation of findings is necessary.

The finding and results are adequately presented in the current paper. The current content and length of the manuscript is very long (around 42 pages). We have decided to prepare the second part to present more results and technical discussion.

The conclusion has no sign of outputs. The results should be summarized and explained somehow in a quantified view. It should totally revise.

Thanks to the comments, the relevant text is added in the conclusion part.

The whole manuscript converted to PDF in the wrong way. lots of paragraphs moved, some links removed and some figures mixed. Editorial revision is necessary.

Thanks to the comments, it was addressed.
